# Path Planning of Unmanned Aerial Vehicles Based on an Improved Bio-Inspired Tuna Swarm Optimization Algorithm

**DOI:** 10.3390/biomimetics9070388

**Published:** 2024-06-26

**Authors:** Qinyong Wang, Minghai Xu, Zhongyi Hu

**Affiliations:** 1School of Artificial Intelligence, Zhejiang College of Security Technology, Wenzhou 325016, China; 2School of Intelligent Manufacturing and Electronic Engineering, Wenzhou University of Technology, Wenzhou 325035, China; 3Institute of Intelligent Information System, Wenzhou University, Wenzhou 325000, China

**Keywords:** UAV, 3D path planning, tuna swarm optimization, horizontal crossover strategy, Levy flight

## Abstract

The Sine–Levy tuna swarm optimization (SLTSO) algorithm is a novel method based on the sine strategy and Levy flight guidance. It is presented as a solution to the shortcomings of the tuna swarm optimization (TSO) algorithm, which include its tendency to reach local optima and limited capacity to search worldwide. This algorithm updates locations using the Levy flight technique and greedy approach and generates initial solutions using an elite reverse learning process. Additionally, it offers an individual location optimization method called golden sine, which enhances the algorithm’s capacity to explore widely and steer clear of local optima. To plan UAV flight paths safely and effectively in complex obstacle environments, the SLTSO algorithm considers constraints such as geographic and airspace obstacles, along with performance metrics like flight environment, flight space, flight distance, angle, altitude, and threat levels. The effectiveness of the algorithm is verified by simulation and the creation of a path planning model. Experimental results show that the SLTSO algorithm displays faster convergence rates, better optimization precision, shorter and smoother paths, and concomitant reduction in energy usage. A drone can now map its route far more effectively thanks to these improvements. Consequently, the proposed SLTSO algorithm demonstrates both efficacy and superiority in UAV route planning applications.

## 1. Introduction

As unmanned aerial technology develops, unmanned aerial vehicles (UAVs) have garnered significant attention because of their ability to function in hazardous and complex environments [1]. Path planning and design, which entails choosing a safe, realistic, and effective flight path under certain constraints from a starting point to a destination, is one of the most crucial aspects of UAV mission systems [2,3]. Because of this, it is possible to see the UAV route planning problem as a difficult optimization problem that requires effective solutions [4,5]. Several solutions have been proposed by researchers to overcome UAV route planning challenges. In addition to more modern techniques like artificial potential field algorithms, they also include more traditional techniques like neural network algorithms [6], Q-learning algorithms [7], rapidly exploring random trees (RRTs) [8], and artificial potential field algorithms. However, these methods usually demand a large time and computational resource commitment. The stochastic search algorithms that underpin swarm optimization techniques are derived from natural occurrences and biological intelligence. Their fundamental idea is to abstract collective behaviors in nature, including foraging and reproduction among particular species, and then quantify these features to construct mathematical models that may be applied to a wide range of problems. Numerous clever optimization algorithms have been created, greatly enhancing optimization techniques and enabling the resolution of combinatorial optimization problems that are challenging to resolve using conventional techniques. They also provide a novel way to explore the concepts and mechanisms of the biological world from many perspectives.

Swarm intelligence optimization methods compensate for the shortcomings of conventional optimization approaches, enabling them to address complex real-world situations successfully [9,10,11]. As a result, more scholars are drawing inspiration for their own swarm intelligence optimization techniques from a variety of biological phenomena [12,13,14]. These algorithms are well known for their ease of use and efficacy in handling a broad variety of real-world applications. They are employed to handle problems related to function optimization [15]. They draw inspiration from natural phenomena or group actions. They offer fresh viewpoints on how to approach difficult path planning issues in unmanned aerial systems. A collaborative coevolutionary spider monkey optimization technique was proposed by Zhu et al. [16] and used for obstacle avoidance in unmanned combat aerial vehicle route planning. These metaheuristic algorithms perform better when dealing with complex environment path planning challenges. Shi et al. introduced an improved adaptive grey wolf optimization (AGWO) method in [17] specifically designed to solve three-dimensional path planning issues for unmanned aerial vehicles (UAVs) operating in complicated settings. All things considered, these metaheuristic algorithms seem to be useful approaches for dealing with difficult path planning problems in UAV applications.

In recent years, these algorithms have been widely applied to handle difficult optimization problems such as path planning, picture processing, and production scheduling. Chinese researchers Xie et al. introduced the novel swarm intelligence optimization approach known as tuna swarm optimization (TSO) in 2021. It was influenced by the helical and parabolic foraging patterns seen in tuna schools [18]. The advantages of TSO’s deployment, simplicity, and ease of understanding have sped up its acceptance. Although the TSO approach is relevant to UAV route planning issues, it has limitations, such as a slow convergence time, susceptibility to local optima entrapment, and instability concerns [19]. Specifically developed for UAV route planning in hazardous and complicated situations, this work proposes a sinusoidal and Levy flight-guided tuna swarm optimization (SLTSO) method that blends the golden ratio sinusoidal approach and Levy flight strategy to overcome these shortcomings. The fundamental component of the SLTSO algorithm is the use of sinusoidal strategies to update individual positions within the population. By utilizing nonlinearly decreasing search factors and weight factors, the sine–cosine algorithm is improved while maintaining a balance between local exploitation and global exploration. The algorithm then uses the Levy flight mechanism to skew the follower’s location updates in relation to the best solution, broadening the search space and raising the possibility of breaking out of local optima. Through comparisons against numerous basic intelligent algorithms and other swarm intelligence optimization algorithms, as well as comparative assessments utilizing several performance metrics on benchmark test functions, the efficacy and superiority of SLTSO are established. Furthermore, the application of SLTSO to engineering optimization design challenges validates its benefits and resilience.

This study proposes an improved tuna optimization algorithm (SLTSO) based on the sine and Levy flight strategy methods. The main contributions of the research can be summed up as follows:(1)A mathematical model was established by reviewing existing drone path optimization strategies.(2)A new unmanned aerial vehicle path optimization strategy was created using an improved tuna optimization algorithm, and it was improved through sine and Levy’s flight.(3)The simulation results were used to confirm the efficiency and effectiveness of the proposed technology.(4)The performance indicators of the proposed algorithm were compared with seven other algorithms.

## 2. Relate Works

Unmanned aerial vehicles (UAVs) are becoming more and more common in science and technology because of their many benefits, which include their affordability, simplicity of use in dangerous environments, tiny size, and excellent mobility. Some initial applications in fields including crisis management, target tracking, and military surveillance have already resulted from these benefits [20]. In order to enable UAVs to avoid obstacles safely while completing flight missions, a major focus of UAV path planning research is to investigate practical path planning models and solution algorithms that take into account variables like flight distance, altitude, turning angles, and obstacle threats [21]. Route planning models and solution algorithms are the two main areas of research for UAV route planning. Building optimization models based on parameters such as path length, flight height, angles of flight, flight duration, and obstacle threats is the primary focus of model research endeavors. Additionally, the computing efficiency of these models has been investigated [22].

Intelligent optimization and numerical optimization are the two primary kinds of algorithms utilized today to solve UAV route planning models [23]. Accurate iterative approaches like the previous one include conventional methods such as visibility graph (VG), artificial potential field (APF), and rapidly exploring random trees (RRTs). In simple scenarios, these techniques may be able to address UAV route planning problems because of their great optimization efficiency [24]. However, when dealing with three-dimensional path planning problems under varying obstacle threat conditions, these models’ high number of path nodes and significant computational loads make it difficult to produce flight route solutions that satisfy all constraint requirements [25]. This is a result of restrictions in both the UAV and the outside environment. For example, Kelner et al. [26] used the rapidly exploring random tree algorithm to solve UAV path planning problems. They treated the flight path’s beginning as the search tree’s root node and chose nearby nodes that met both constraint limits and had the lowest possible cost to include in the search tree. They repeated this process until they were able to determine a UAV flight route. However, the algorithm has difficulty locating exits among obstacle clusters when there are a lot of closely spaced barriers and very few escape pathways. As a consequence, the global best path cannot be found. For static UAV path planning, Cao et al. [27] used the artificial potential field approach, which is simple to use and has a quick convergence time but produces flight route options that are of lower quality. Furthermore, the produced routes have a tendency to approach concave-shaped barriers in scenarios with several complicated obstacles, which might result in the failure of the planned flight paths. Under the assumption that the locations of three-dimensional obstacles were known, Blasi et al. [28] used the visibility graph approach with flight path length as a performance parameter to create path planning routes for rotorcraft UAVs. While this approach offers a solution, it is not appropriate for locations with a high number of obstacles, and the resulting flight path plans frequently result in low flying efficiency. All things considered, even if numerical optimization methods have worked well in some situations, they have drawbacks in complex or severely limited environments. Therefore, there is a constant need for improvements in intelligent optimization algorithms designed to overcome these obstacles and improve UAV path planning’s flexibility and efficiency.

In the study of UAV route planning, swarm intelligent optimization—a metaheuristic random search algorithm—has been very important. Its high chance of obtaining a globally optimal solution stems from its directed and random character in the solution process, quick speed of optimization, and steady quality of solution search [29]. A spherical vector-based particle swarm optimization (PSO) technique, for instance, was presented by Phung et al. [30], which takes into account variables including UAV route length, altitude, angles, and obstacle threats. This approach efficiently shrinks the search space, improving the quality of the solutions sought and flight safety by utilizing the magnitude, elevation, azimuth components, and turn and climb angles of linked vectors. By dividing the UAV flight time evenly using a timestamp segmentation technique and adding social hierarchy concepts to the fundamental pigeon-inspired optimization (PIO) algorithm, Wang et al. [31] created an enhanced PIO algorithm that can successfully solve multi-UAV cooperative path planning problems. These algorithms have a straightforward structure, speedy optimization, and the ability to produce excellent path planning schemes that adhere to UAV flying parameters. Though these benefits make UAV route planning issues easier to solve, they struggle to provide sufficient practicable flight path plans because they are sluggish in developing workable solutions that avoid obstacles and frequently become stuck in local searches.

To sum up, the majority of the research performed on UAV path planning models at the moment is focused on building flight route planning models that are constrained by ground impediments and performance metrics like flight duration and distance. Planning models that incorporate multiple performance indicators and handle intricate obstacle settings are comparatively less often designed. This is especially true for models that incorporate ground and airspace limits as well as multi-performance indicator optimization. As a result, new or enhanced intelligent optimization techniques are set to replace conventional approaches as the go-to solutions for UAV route planning issues. In order to do this, this work introduces a unique approach to UAV route planning that makes use of the Levy flight-guided tuna swarm optimization (SLTSO) algorithm in conjunction with a sinusoidal strategy.

## 3. Mathematical Model for Path Planning of UAVs

Unmanned aerial vehicle (UAV) route planning model design takes into account factors including the flight path, the airspace that may be flown through, and flight characteristics like altitude, angle, threat level, and flight distance, which are all used as performance indicators [32].

### 3.1. The Problem with Trajectory Planning

An essential component of a UAV’s capacity to finish a mission is trajectory planning, which has a direct impact on the effectiveness and efficiency of task execution [33]. Trajectory planning is the process of selecting, among a large number of viable paths, an ideal or nearly ideal path that avoids obstructions and satisfies predetermined requirements, as shown in Figure 1.

### 3.2. Path Encoding

The path of a UAV can be regarded as a series of line segments connected by coordinate points. Therefore, the path is encoded using a real number encoding method, which is represented as follows:(1)$$\left(x_{1},x_{2},\ldots ,x_{n},y_{1},y_{2},\ldots ,y_{n},z_{1},z_{2},\ldots ,z_{n}\right)$$
where $x_{i}(i=1,2,\cdots ,n)$ represents the *x*-axis coordinate of the *i*-th trajectory point. The parameters $y_{i}(i=1,2,\cdots ,n)$ and $z_{i}(i=1,2,\cdots ,n)$ represent the *y*-axis and *z*-axis coordinates of the *i*-th trajectory point. The parameter *n* is the number of path nodes.

### 3.3. Constraint Condition

The path planning model of unmanned aerial vehicles includes terrain constraints and corner constraints. The terrain constraints are represented as follows:(2)$$h\left(x_{i},y_{i}\right)+z_{\min}<z_{i}<z_{\max},i=1,2,\ldots ,n$$
where $h\left(x_{i},y_{i}\right)$ represents the terrain height corresponding to the two-dimensional terrain coordinates. *Z_min_* and *Z_max_* represent the minimum and maximum safe flight altitude of the UAV, respectively. Other constraints, such as minimum track segment length, maximum track length, and maximum turning angle, can be expressed as follows [34]:(3)$$\theta_{i}⩽\theta_{\max},i=1,2,\ldots ,n$$
(4)$$L_{i}⩾L_{\min},i=1,2,\ldots ,n$$
(5)$$\sum_{i=1}^{n}L_{i}⩽L_{\max},i=1,2,\ldots ,n$$
where the parameter $\theta_{i}$ represents the turning angle of the *i*-th trajectory point. $\theta_{\max}$ is the maximum turning angle. $L_{i}$ is the path length for the *i*-th segment of the trajectory. *L_min_* is the minimum track segment length, and *L_max_* is the maximum track length.

### 3.4. Optimization Objectives

In this article, fuel consumption and risk are considered as two optimization objectives. Among them, these two goals may conflict with each other in some cases. For example, the shorter the path, the greater the threat.

#### 3.4.1. Fuel Consumption

Generally speaking, a shorter path length means shorter flight time and also requires less fuel consumption. In this article, fuel consumption refers to the amount of fuel consumed by a UAV during its flight from the starting point to the target point during mission execution. Let $f_{\mathrm{fuxl}}$ be the fuel consumption cost. If the speed of the UAV remains constant during flight, fuel consumption is directly proportional to the length of the flight path. Therefore, fuel consumption $f_{\mathrm{fuxl}}$ can be calculated as follows:(6)$$f_{\mathrm{fuxl}}=\alpha^{*}\sum_{i=1}^{n}L_{i}=\alpha^{*}\sum_{i=1}^{n}\sqrt{{\left(x_{i}-x_{i-1}\right)}^{2}+{\left(y_{i}-y_{i-1}\right)}^{2}+{\left(z_{i}-z_{i-1}\right)}^{2}}$$
In Equation (6), the parameter α is the fuel coefficient and *L_i_* is the path length of the *i*-th segment of the trajectory.

#### 3.4.2. Threats

This study mainly considers detecting threats and high threats. Assuming there are *m* detection threats, the projection center coordinates of the *k*-th (*k* = 1, 2, …, *m*)) detection threat are *C_k_*. The threat radius is *R_k_*. When the UAV flies from path node *W_i_* to node *W_i_*_+1_, the straight-line distance from *C_k_* to *W_i_ W_i_*_+1_ is *d_k_*, as shown in Figure 2.

At this point, the detection threat value of the UAV is inversely proportional to its distance *d_k_*. Therefore, the detection threat $f_{\mathrm{detect}}$ can be calculated as follows:(7)$$f_{\mathrm{detect}}=\sum_{i=1}^{n}\sum_{k=1}^{m}T_{i,k}$$
(8)$$T_{i,k}=\left\{\begin{array}{l} 0,d_{k}⩾R_{k}\\ R_{k}/d_{k},0<d_{k}<R_{k}\\ \infty ,\;d_{k}=0 \end{array}\right.$$
In Equation (8), $T_{i,k}$ represents the detection threat value from the *k*-th interference center received by the UAV on the *i*-th trajectory.

The higher the flying altitude of a UAV, the greater the probability of being detected by an enemy. Therefore, flight altitude is also one of the threats that this article needs to consider. In a 3D map, UAVs should strive to maintain an average safe altitude while preventing collisions with the terrain environment. Therefore, the high threat $f_{\mathrm{altitude}}$ can be calculated as follows:(9)$$f_{\mathrm{altitude}}=\sum_{i=1}^{n}H_{i}$$
(10)$$H_{i}=\left\{\begin{array}{cc} \left|z_{i}-\frac{z_{\max\;}+z_{\min\;}}{2}\right| & ,h\left(x_{i},y_{i}\right)+z_{\min\;}<z_{i}<z_{\max\;}\\ \infty & ,\;\mathrm{others}\; \end{array}\right.$$

In Equation (10), the parameter *H_i_* represents the altitude threat of the UAV in the *i*-th trajectory. Therefore, the threat of UAVs $f_{\mathrm{threat}}$ can be expressed as
(11)$$f_{\mathrm{threat}}=f_{\mathrm{dctect}}+f_{\mathrm{altitude}}$$

### 3.5. Path Planning Model for UAVs

According to Section 3.2 and Section 3.3, the three-dimensional path planning model for unmanned aerial vehicles can be represented as:(12)$${\left.J=\left[\min\left(f_{\mathrm{fuel}}\right),\min\left(f_{\mathrm{threat}}\right)\right]\right)}^{T}$$
(13)$$s.t\;\left\{\begin{array}{l} h\left(x_{i},y_{i}\right)+z_{\min}<z_{i}<z_{\max},i=1,2,\ldots ,n\\ \theta_{i}⩽\theta_{\max},i=1,2,\ldots n\\ L_{i}⩾L_{\min},i=1,2,\ldots n\\ \sum_{i=1}^{n}L_{i}⩾L_{\max},i=1,2,\ldots n \end{array}\right.$$
In Equation (12), **J** is the objective vector, and the problem is a constrained minimization bi-objective optimization problem.

## 4. Tuna Swarm Optimization Algorithm (TSO)

In 2021, researcher Lei Xie presented the population-based global optimization metaheuristic method known as “tuna swarm optimization” (TSO). The TSO algorithm imitates two foraging strategies found in schools of tuna. The first, known as spiral foraging, explains how the fish swim in a spiral pattern to guide prey toward shallow waters where they can feed; the second, known as parabolic foraging, involves a fish swimming in a parabolic configuration, trailing the tail of the fish in front of it to encircle the school of tuna. TSO achieves global optimization by using these two foraging strategies [35]. The Tuna swarm optimization method is a unique population-based metaheuristic algorithm that mimics the helical and parabolic feeding patterns of tuna schools. It was first presented by researchers such as Xie et al. in 2021 [36]. This approach is distinguished by its robust exploratory capabilities and fewer configurable parameters. Currently, the tuna algorithm finds use in many different fields. Fan et al. [37] used an enhanced tuna swarm method to tackle the economic operation problem in hydropower facilities. A new and enhanced tuna optimization particle filter technique was suggested by Fu et al. [38], which effectively resolves the power system harmonic estimation problem. An improved tuna swarm approach was used by Gou et al. [39] to address Jensen model parameter estimation issues. Nanda et al. [40] used an ELM model based on an enhanced tuna swarm method to maximize the prediction of solar power production. An adaptive black hole–tuna swarm technique was introduced by Sheeja et al. [41] for multi-objective energy conservation optimization in wireless sensor networks.

### Standard Tuna Swarm Optimization Algorithm (TSO)

The underlying principle of the tuna swarm optimization algorithm (TSO) is that every member of the population is figuratively represented as a tuna and that each tuna uses its own foraging strategy to search for the best solutions while being impacted by the other tunas’ foraging activities [42]. Every tuna modifies its location according to its own fitness and the fitness of other tunas in each algorithm iteration, improving its environmental adaptation and pursuing the global optimum [43].

(1)Population Initialization

In the standard tuna swarm algorithm, the initialization of the population follows a similar approach to most population-based metaheuristics, where an initial population is randomly generated within the search space. The mathematical expression for this process can be denoted as Equation (14):(14)$$p_{i}^{n}=\mathrm{rand}\cdot (ub-lb)+lb,i=1,2,3,\ldots ,NP$$
where *rand* is a random number in the interval [0, 1], *ub* is the upper limit of the search space, *lb* is the lower limit of the search space, $p_{i}^{n}$ represents the initialization value of the tuna population, and *NP* is the number of populations.

(2)Tuna school spiral feeding

The first foraging strategy for tuna schools is spiral foraging. When tunas feed, they form a spiral shape that drives them to shallow water areas where they are more vulnerable to attack. The mathematical model of the spiral foraging strategy is as follows:(15)$$X_{i}^{t+1}=\left\{\begin{array}{l} \left\{\begin{array}{l} \alpha_{1}\cdot \left(X_{\mathrm{best}}^{t}+\beta \cdot \left|X_{\mathrm{best}}^{t}-X_{i}^{t}\right|\right)+\alpha_{2}\cdot X_{i}^{t},i=1\\ \alpha_{1}\cdot \left(X_{\mathrm{best}}^{t}+\beta \cdot \left|X_{\mathrm{best}}^{t}-X_{i}^{t}\right|\right)+\alpha_{2}\cdot X_{i-1}^{t},i=2,3,\ldots ,NP \end{array}\;\;\;\mathrm{rand}\;<\frac{t}{t_{\max}}\;(15\text{-}a)\right.\\ \left\{\begin{array}{l} \alpha_{1}\cdot \left(X_{\mathrm{rand}}^{t}+\beta \cdot \left|X_{\mathrm{rand}\;}^{t}-X_{i}^{t}\right|\right)+\alpha_{2}\cdot X_{i}^{t},i=1\\ \alpha_{1}\cdot \left(X_{\mathrm{rand}\;}^{t}+\beta \cdot \left|X_{\mathrm{rand}\;}^{t}-X_{i}^{t}\right|\right)+\alpha_{2}\cdot X_{i-1}^{t},i=2,3,\ldots ,NP \end{array}\;\;\mathrm{rand}\;<\frac{t}{t_{\max}}\;(15\text{-}b)\right. \end{array}\right.$$
(16)$$\alpha_{1}=a+(1-a)\cdot \frac{t}{t_{\max}}$$
(17)$$\alpha_{2}=(1-a)-(1-a)\cdot \frac{t}{t_{\max}}$$
(18)$$\beta =e^{bl}\cdot \cos(2\pi b)$$
(19)$$l=e^{3\cos\left(\left(\left(t_{\max}+1/t\right)-1\right)\pi \right)}$$

In the above equation, $X_{i}^{t+1}$ represents the individual position in the *t* + 1 iteration. $X_{\mathrm{best}\;}^{t}$ represents the optimal position of the current individual. $X_{\mathrm{rand}\;}^{t}$ represents the randomly generated individual positions in the search space. $\alpha_{1}$ and $\alpha_{2}$ are weight coefficients that control the tendency of an individual to move towards the optimal individual and the previous individual, respectively. The parameter *a* is a constant and represents the degree coefficient of the tuna following the optimal individual and the previous individual in the initial stage. The parameter *t* represents the current number of iterations. The parameter $t_{\max}$ represents the maximum number of iterations. The parameters *β* and *l* are intermediate variables. The parameters *b* and *rand* are represented as random numbers uniformly distributed between [0, 1]. The parameter *NP* represents the tuna population size. Tunas enhance their search capability in the surrounding space through spiral foraging behavior, as per Equation (15-a), where, when all tunas spiral forage around food sources, they exhibit a strong capacity to explore the search space effectively to locate optimal solutions. However, when the best-performing individual fails to find food, blindly following the optimal individual may lead to decreased search efficiency for the entire group. Consequently, to augment the global search ability of the tuna swarm algorithm, a random individual position is introduced as a reference point for spiral searching, according to Equation (15-b). This helps each tuna to probe a broader area in its quest. As the number of algorithm iterations increases, the tuna swarm algorithm gradually shifts the reference point for spiral foraging from a randomly chosen individual’s location to that of the optimal individual, thereby enhancing local search capability and search precision [44].

(3)Tuna parabolic foraging

The second tuna feeding strategy is parabolic feeding, where each tuna swims along with the previous one, forming a parabolic shape to surround the tuna. In addition, tunas also search for food around themselves. The mathematical model of the parabolic foraging strategy is as follows:(20)$$X_{i}^{t+1}=\left\{\begin{array}{ll} X_{\mathrm{best}}^{t}+rand\;\cdot \left(X_{\mathrm{rand}}^{t}-X_{i}^{t}\right)+TF\cdot p^{2}\cdot \left(X_{\mathrm{rand}}^{t}-X_{i}^{t}\right) & \;rand<0.5\\ TF\cdot p^{2}\cdot X_{i}^{t} & \;rand\;\geq 0.5 \end{array}\right.$$
(21)$$p={\left(1-\frac{t}{t_{\max}}\right)}^{\left(\frac{t}{t_{\max}}\right)}$$
where *TF* in Equation (20) is a random number of −1 or 1, used to control the direction of population development, and the parameter *p* is an adjustment factor used to control the magnitude of population development [45].

Tunas cooperate through the above two foraging strategies, continuously updating their individual positions until the termination conditions are met. In each iteration, each tuna updates its individual historical optimal position and global historical optimal position based on its own position and fitness value and finally returns the optimal individual position and its corresponding fitness value.

## 5. Improved TSO Algorithm Based on the Sine Strategy and Levy Flight (SLTSO)

### 5.1. Elite Opposition-Based Learning Mechanism

It has been found via scientific study that applying a reverse learning process can lead to a 50% improvement in the likelihood of finding the ideal answer. The fundamental idea behind this concept is to create a new population by choosing the best individuals from both possible solutions and their inverses [46]. Building on this basis, the elite opposition-based learning (EOBL) approach creates a reverse population that is exclusive to the elite members of the original population. Because elite people are more likely to possess relevant knowledge, EOBL increases population quality overall and search efficiency more than standard opposition-based learning [47]. The EOBL method enhances the likelihood of locating the global optimum by concentrating on elite people, which successfully keeps the population from being stuck in local optima during the optimization process. By focusing computing resources on high-performing regions, this selective strategy optimizes the trade-off between exploration and exploitation while preserving the possibility to explore beyond the current best solutions.

The definition of elite reverse solving assumes that $X_{i,j}^{E}=\left(X_{i,1}^{E},X_{i,2}^{E},\cdots ,X_{i,d}^{E}\right)$,(i=1,2,⋯,N),(j=1,2,⋯,d), is an elite individual in *d*-dimensional space, and its inverse solution is defined as $\bar{X_{i,j}^{E}}=\left(X_{i,1}^{E},X_{i,2}^{E},\cdots ,X_{i,d}^{E}\right)$, such that
(22)$$\bar{X_{i,j}^{E}}=c\cdot \left({\mathrm{lb}}_{j}+{\mathrm{ub}}_{j}\right)-X_{i,j}^{E}$$
where $X_{i,j}^{E}\in \left[lb_{j},ub_{j}\right],c\in [0,1]$, the parameter *c* is a random number between 0 and 1, and $ub_{j}=\max(X_{i,j})$ and $lb_{j}=\min(X_{i,j})$ are the upper and lower bounds of the dynamic search boundary. Dynamic changes in search boundaries can save previous optimization experience and improve search efficiency. Individuals beyond the search boundary are reset using Formula (23):(23)$$\bar{X_{i,j}^{E}}=\mathrm{rand}\left(lb_{j}+ub_{j}\right)$$

When using the elite reverse learning strategy to calculate the reverse solution for the initial population and the elite individuals in the later iteration stage, the reverse solution for the top *N*/2 individuals in the population fitness ranking is calculated according to Equation (22). The reverse population and the original population are sorted based on individual fitness, and the optimal *N* elite individuals are selected to form the next generation of optimization population. This strategy can effectively avoid blind searching by algorithms and reduced solution efficiency, laying the foundation for improving solution accuracy and accelerating convergence speed.

### 5.2. Introducing the Levy Flight Strategy and the Greedy Strategy to Update Positions

Levy flight is used to update the position of a tuna in the TSO algorithm. Levy flight is a random flight with alternating search ranges, which can enhance the algorithm’s global search ability, making it possible for the algorithm to jump out of local optima [48,49]. The mathematical expression for position update is shown in Equation (24):(24)$$xl_{i}(t)=x_{i}(t)+l\times C1⊗levy$$

In Formula (24), $xl_{i}(t)$ represents the position of the Levy flight update, and the parameter *1* is the step weight. Levy represents the step size that follows the Levy distribution, and the step size calculation formula is shown in Equation (25):(25)$$\;\mathrm{levy}\;=u/|v{|}^{\frac{l}{\beta_{l}}}$$
where the parameters *u* and *v* follow a normal distribution, $u\sim N\left(0,\sigma_{u}^{2}\right),v\sim N\left(0,\sigma_{v}^{2}\right)$. The definitions of $\sigma_{u}^{2}$ and $\sigma_{v}^{2}$ are shown in Equations (26) and (27).
(26)$$\sigma_{u}={\left[\frac{\Gamma (1+\delta )\sin(\delta \pi /2)}{\Gamma ((1+\delta )/2)2^{(\delta -1)/2}\delta }\right]}^{1/\delta }$$
(27)$$\sigma_{v}=1$$
where the parameter δ takes a value of 1.5.

After updating the positions of tunas using Levy flight, because of the random flight characteristics of Levy flight, the updated position of a may not necessarily be a better position. Therefore, a greedy strategy is adopted to choose whether to retain the original position or find a new position. The selection formula is shown in Equation (28).
(28)$$x_{i}(t)=\left\{\begin{array}{l} x_{i}(t),\;\mathrm{fitness}\;\left(x_{i}(t)\right)\leq \;\mathrm{fitness}\;\left(xl_{i}(t)\right)\\ xl_{i}(t),\;\mathrm{fitness}\;\left(x_{i}(t)\right)>\;\mathrm{fitness}\;\left(xl_{i}(t)\right) \end{array}\right.$$
where, in Formula (28), *fitness*(*x*(*t*)) and *fitness*(*xl*(*t*)) respectively represent the original tuna position calculation solution of the algorithm and the tuna position calculation solution after the Levy flight update. To distinguish the updated position for a tuna, $x_{i}(t)$ is assigned to $xl_{i}(t)$ in Equation (28), thus obtaining Equation (29):(29)$$Xh_{i}(t)=x_{i}(t)$$

### 5.3. Introducing the Golden Sine Strategy to Update Individual Positions

This work also offers uses the golden sine approach to update a tuna’s position in response to the sluggish convergence speed of TSO and its tendency to easily fall into local optima. The golden sine algorithm is the source of the golden sine strategy [50]. First, because of the strong global search ability of this method, which is based on the angle relationship between the unit circle and the sine function, the sine function may traverse all points within the search range of the unit circle [51]. Second, the algorithm’s convergence speed is accelerated by employing the golden ratio coefficient to condense the solution space and provide a possibly ideal search region [52].

The position update formula used in the golden sine strategy to update the position of a tuna in the previous iteration is shown in Equation (30):(30)$$xg_{i}(t)=x_{i}(t)\left|\sin\left(r_{g1}\right)\right|+r_{g2}\sin\left(r_{g2}\right)\left|c_{1}T_{\mathrm{pos}\;}-c_{2}x_{i}(t)\right|$$
where $xg_{i}(t)$ is the tuna position updated by the golden sine strategy, $r_{g1}$ is a random number within [0, 2π], $r_{g2}$ is a random number within [0, π], and *c*_1_ and *c*_2_ are the golden ratio coefficients. The expressions for *c*_1_ and *c*_2_ are shown in Equations (31) and (32):(31)$$c_{1}=a\tau +b(1-\tau )$$
(32)$$c_{2}=a(1-\tau )+b\tau$$
where τ = ($\sqrt{5}$ − 1)/2~0.6183. The value of the parameter *a* is −π, and the value of the parameter *b* is π.

Then, $x_{i}(t)$ in the tuna position update formula is replaced with $xg_{i}(t)$, thus reducing the optimization space, accelerating the tuna search speed, and improving the algorithm convergence speed. The new tuna position update formula is shown in Equation (33):(33)$$x_{i}(t+1)=T_{pos}+C1Z\cos\left(2\pi R_{4}\right)\times \left(T_{pos}-xg_{i}(t)\right)$$
where *R*_4_ is a random number within the range of [−1, 1]. To distinguish the updated position for a tuna, $x_{i}(t)$ is assigned to $Xp_{i}(t)$ in Equation (33), and the formula becomes Equation (34).
(34)$$Xp_{i}(t)=x_{i}(t)$$
Combining the optimized tuna and individual search agent mechanisms, the final algorithm position update formula is obtained by combining Equations (29) and (34) to obtain Equation (35). *R*_5_ is a random number within the range of [0, 1], and *β* is the adjustment parameter.
(35)$$\begin{array}{l} x_{i}(t)=Xh_{i}(t),R_{5}<\beta \;(35\text{-}a),\\ x_{i}(t)=Xp_{i}(t),R_{5}\geq \beta \;(35\text{-}b) \end{array}$$

### 5.4. The Workflow for the Proposed SLTSO Algorithm

Based on the above methods, the standard tuna swarm algorithm was improved. The steps for applying the improved SLTSO algorithm to UAV trajectory planning are as follows:

Step 1: Initialize the basic parameters of the improved tuna swarm algorithm. The population size is *NP*, the maximum number of iterations is *t_max_*, the position vector dimension is *D*, the upper and lower limits of the search space are *ub* and *lb*, and the probability parameter is *z*.

Step 2: Use the elite reverse learning mechanism to initialize the position of the tuna school.

Step 3: Calculate the fitness value *J* of a tuna according to Equation (12), and determine whether the constraint conditions are met using Equation (13). Compare the fitness values that meet the constraint conditions, update the optimal value of the current individual tuna (i.e., the fitness value of the tuna), and save it.

Step 4: When the algorithm enters the iterative process, if the random number rand generated in [0, 1] is less than *z*, update the position of the tuna according to Equation (14) and proceed to step 8.

Step 5: If the random number *rand* generated within [0, 1] satisfies the condition rand ≥ z and also meets the condition *p* < *p*′, and if another random number rand generated within [0, 1] is less than *t*/*t_max_*, update the tuna’s position according to Equation (15-a) and proceed to Step 8.

Step 6: If the condition *p* < *p*″ is met and the random number *rand* is generated in [0, 1], and rand ≥ *t*/*t_max_*, update the position of the tuna according to Equation (28) and proceed to step 8.

Step 7: If the condition *p* ≥ *p*′ is met, update the position of the tuna according to Equation (20) and proceed to step 8.

Step 8: Calculate the fitness value of the updated tuna position according to Formula (30), compare the fitness value that meets the constraints of the UAV with the optimal value of the previously saved individual tuna, and update the compared optimal value to the current individual tuna’s optimal value and save it.

Step 9: Determine if the maximum number of iterations has been reached. If it has, stop the iteration and output the optimal fitness value of the tuna individual. Otherwise, return to step 4.

The workflow of the SLTSO algorithm is shown in Figure 3.

### 5.5. Time Complexity Analysis of the SLTSO Algorithm

Time complexity is a crucial metric for evaluating the performance of an algorithm. It assumes the size of the tuna swarm population is *N*, the dimensionality of the problem is *D*, and the maximum number of iterations is *T*, with the time required to evaluate the objective fitness function denoted as *f*(*D*). According to the implementation steps of the SLTSO (Sine–Levy truncated tuna swarm optimization) algorithm, the initialization phase has a time complexity of *O*(*N* · (*f*(*D*) + *D*)), meaning that the time required scales linearly with the population size and the combined effort of evaluating the fitness function and initializing the dimensions.

Since the tuna optimization strategy in the SLTSO algorithm modifies only the update method for discoverers without adding extra computational steps, the position update stage has a time complexity of *O*(*N* · *D*), indicating that the position update operations for each individual take place independently and proportionally to the number of dimensions. The phase of applying position update strategies also incurs a time complexity of *O*(*N* · *D*). The introduction of the golden sine strategy and Levy flight adds another layer of complexity, which amounts to *O*(*N* · *f*(*D*) + *D*), reflecting the additional overhead for applying these advanced strategies to each individual in the population. Taking all these stages into account, the overall time complexity of the SLTSO algorithm is *O*(*N* · *D* · *T*), which is equivalent to the basic TSO algorithm. Hence, SLTSO does not introduce any additional computational burden in terms of its time complexity compared to the original TSO algorithm.

## 6. Simulation Experiments and Result Analysis

By contrasting it with five sample dynamic multi-objective algorithms, the suggested SLTSO algorithm’s performance is validated and analyzed. The test functions, comparison algorithms, parameter configurations, and analysis of the experimental data are described in detail below. The 12th Generation Intel(R) Core(TM) i9-12900K CPU, running at 3.20 GHz, 64 GB of RAM, and MATLAB software version R2022b were installed on the laptop used in this research. In order to validate the benefits of the refined approach put forward in this work, simulations were run using a population size of 30 and a maximum iteration count of 1000. The standard particle swarm optimization (PSO) [53], dung beetle optimization (DBO) [54], slime mold algorithm (SMA) [55], Harris hawk optimization (HHO) [56], subtraction–averaging-based optimization (SABO) [57], sand cat swarm optimization (SCSO) [58], basic tuna swarm optimization (TSO), and the improved TSO algorithm incorporating both the sine strategy and Levy flight were the algorithms used to benchmark the performance of the SLTSO algorithm.

### 6.1. Comparison of Test Function Results

For numerical simulations, the tests used 30 benchmark test functions from the CEC2018 suite. The test functions for CEC2018 are a set of functions intended to evaluate the effectiveness of intelligent optimization algorithms. There are 30 functions altogether, which include multimodal, unimodal, hybrid, and composite function types. These functions are designed to assess the effectiveness of optimization algorithms in a variety of scenarios, taking into account various dimensions and degrees of difficulty. On all 30 base test functions, the proposed SLTSO algorithm was evaluated against the PSO, DBO, SMA, HHO, SABO, SCSO, and TSO algorithms separately. Each function was run independently for 50 trials. The objective of this comparison was to examine the optimization performance and stability benefits of the SLTSO algorithm over alternative swarm intelligence algorithms. Figure 4 displays the convergence curves of the iterative procedures for various test functions. It is noteworthy that Function F2 encountered difficulties during execution, which were caused by issues with the CEC 2018 test function itself. As a result, no results were provided for this specific function.

In this study, convergence curves based on iteration counts and fitness values for the test functions are presented to give a more comprehensible comparison of the convergence accuracy and speed of the methods. The optimization performance of the proposed SLTSO method, when compared to the PSO, DBO, SMA, HHO, SABO, SCSO, and TSO algorithms, is typically similar, as shown in Figure 4 for Functions F1 through F5. However, certain specific situations may not perform as well as optimally. But, in terms of Functions F19 through F30, the SLTSO algorithm performs several orders of magnitude better than the other algorithms. Its standard deviation is also typically lower than the other algorithms’ standard deviations, suggesting that the SLTSO algorithm keeps a higher diversity in its initial population, which greatly improves its stability. With direct search to optimal values and 100% optimization efficiency achieved for Functions F12 and F14, the SLTSO algorithm demonstrates optimization performance for Functions F12 to F18 that is comparable to that of PSO, DBO, SMA, HHO, SABO, SCSO, and TSO. This is a significant improvement over the unmodified PSO, DBO, SMA, HHO, SABO, and SCSO algorithms. This implies that the SLTSO’s implementation of the Levy flight optimization strategy and sine approach promotes population diversity, expands the search space, and strengthens the searcher’s exploration technique through individual position updates, enabling the algorithm to escape local optima successfully.

The convergence curves of the test functions show that, in general, the SLTSO method has a faster rate of convergence than the PSO, DBO, SMA, HHO, SABO, and SCSO algorithms. Functions F20 through F30 are high-dimensional multi-modal functions that are used to evaluate the global exploration capabilities of algorithms. Because of their global optima, they are difficult to optimize. It is clear from their individual convergence curves that the SLTSO algorithm outperforms the PSO, DBO, SMA, HHO, SABO, SCSO, and TSO algorithms in terms of convergence speed and precision. This suggests that while optimizing high-dimensional functions, the other algorithms commonly become stuck in local optima, significantly slowing down their convergence rates. SLTSO, on the other hand, initially uses an elite reverse learning process to guarantee initial population diversity. The discoverer’s position update technique is then improved by the sine optimization strategy, which keeps the algorithm from being trapped in local optima in the early iterations. Finally, by applying perturbations to the top performers, the Levy flight strategy allows the algorithm to exit local optima quickly and finally converge to the global optimum value.

In conclusion, the SLTSO algorithm shows significant stability and optimization performance improvement for the set of 30 benchmark test functions. It performs especially well for Functions F20 to F30, outperforming the seven other algorithms by a significant margin. Moreover, for high-dimensional multi-modal functions, the SLTSO algorithm has a notable capacity to escape local optima quickly and converge to the global optimum.

For every strategy in this work, the population size and number of iterations are fixed at 30 and 1000, respectively, to improve test result dependability and reduce the impact of randomness inherent in heuristic algorithms. The ideal value findings from the 30 different runs were used to compute statistical variables such as the mean, standard deviation (Std), and min. The algorithm’s average ability to optimize the target function is represented by the mean. For every target function, the min displays the best optimization result out of the thirty executions. The stability of the algorithm’s ability to optimize a given target function is measured by the Std. Table 1, Table 2 and Table 3 present the testing outcomes of the eight optimization algorithms under consideration across 10, 30, and 50 dimensions for the entire set of 30 benchmark functions, providing a comprehensive overview of their comparative performances. These statistical data substantiate the feasibility and effectiveness of the SLTSO algorithm.

Table 1, Table 2 and Table 3 show that the SLTSO algorithm consistently beats the other seven algorithms in terms of mean, Std, and best scores when tested with benchmark Functions F1–F5 across dimensions of 10, 30, and 50. The mean and Std values of the SLTSO algorithm are clearly more than 30% higher than those of the DBO algorithm after a thorough comparison of the mean, Std, and best values. When compared to the other seven algorithms, the SLTSO algorithm performs better overall when taking into account the combined performance on F1 through F5. In the evaluation of benchmark Functions F6-F17, the SLTSO algorithm achieves favorable mean and best parameter values, signifying good performance. On other multi-modal benchmark functions, the SLTSO algorithm surpasses the PSO, DBO, SMA, HHO, SABO, and SCSO algorithms.

Each of the eight methods achieves the theoretically ideal answer for the best value in the mixed benchmark function testing. While all eight methods for the mixed benchmark Functions F12–F16 attain the ideal mean and best values in theory, the SLTSO algorithm performs better on the standard deviation than the other seven techniques. The SLTSO method converges more quickly and needs fewer iterations than the mean and best values of the other algorithms, even if they both approach the theoretical optimum. The SLTSO algorithm outperforms the others in the mean and best scores across all three benchmark function categories. The SLTSO algorithm uses fewer iterations and performs best overall while obtaining the same degree of accuracy.

### 6.2. Comparison of Engineering Application 

To verify the effectiveness and feasibility of the SLTSO algorithm in engineering applications, welding beam design problems and gear system design problems were selected for analysis.

(1)Welding beam design problem

The welding beam design (WBD) problem is a minimization problem in which the optimization algorithm aims to reduce the manufacturing cost of the design [59]. This optimization problem can be described as finding a solution that satisfies the shear stress (*τ*), bending stress (θ), beam bending load (*Pc*), and end deviation (*δ*). The four design variables constrained by boundary conditions, namely, the length (*l*), curvature (*t*), thickness (*b*), and weld thickness (*h*) of the beam, make the cost of manufacturing welded beams the highest. Therefore, the welded beam problem is a typical nonlinear programming problem. The mathematical description of the WBD problem is as follows:(36)$$\mathrm{Variable}:\;\vec{l}=\left[l_{1},l_{2},l_{3},l_{4}\right]=\left[hltb\right]=\left[x_{1}x_{2}x_{3}x_{4}\right]$$
(37)$$\mathrm{Standard}\;\mathrm{function}:f(\vec{l})=1.10471l_{1}^{2}l_{2}+0.04811l_{3}l_{4}(14.0+l_{2})$$
where the objective function represents the cost of manufacturing welded beams, and it is required to solve the problem of cost maximization. The range of decision variable values is as follows: 0.1 ≤ *l*_1_ ≤ 2, 0.1 ≤ *l*_2_ ≤ 10, 0.1 ≤ *l*_3_ ≤ 10, 0.1 ≤ *l*_4_ ≤ 2.

The constraints are $s_{1}(\vec{l})=\tau (\vec{l})-\tau_{\max}\leq 0$, $s_{2}(\vec{l})=\sigma (\vec{l})-\sigma_{\max}\leq 0$, $s_{3}(\vec{l})=\delta (\vec{l})-\delta_{\max}\leq 0$, $s_{4}(\vec{l})=l_{1}-l_{4}\leq 0$, $s_{5}(\vec{l})=0.125-l_{1}\leq 0$. $s_{6}(\vec{l})=1.10471l_{1}^{2}+0.04811l_{3}l_{4}(14.0+l_{2})-5.0\leq 0$,

$s_{7}(\vec{l})=1.10471l_{1}^{2}+0.04811l_{3}l_{4}(14.0+l_{2})-5.0\leq 0$, where *σ_max_* = 30,000 psi, *P* = 6000 lb, *L* = 14 in., δ_max_ = 0.25 in., *E* = 3 × 10^6^ psi, *τ_max_* = 136,000 psi, and G = 1.2 × 10^7^ psi. The expressions of various functions in the constraint conditions refer to Equations (38)–(42).
(38)$$\tau \left(\vec{l}\right)=\sqrt{{\tau \prime }^{2}+2\tau \prime \tau \prime \prime \left(l_{2}/R\right)+{\left(\tau \prime \prime \right)}^{2}}$$
(39)$$\tau =\frac{P}{\sqrt{2}l_{1}l_{2}},\;\tau \prime \prime =\mathrm{MR}/J,\;M=p\left(L+l_{2}/2\right)$$
(40)$$R=\sqrt{\frac{\left[l_{2}^{2}+{\left(l_{1}+l_{3}\right)}^{2}\right]}{4}}$$
(41)$$J=2\left\{\sqrt{2}l_{1}l_{2}[\frac{l_{2}^{2}}{12}+\frac{{\left(l_{1}+l_{3}\right)}^{2}}{14}]\right\}$$
(42)$$P_{c}\left(\vec{l}\right)=\frac{4.013El_{3}l_{4}^{2}}{6L^{2}}\left(1-\frac{l_{3}\sqrt{E}}{8\mathrm{LG}}\right)$$

In this article, the penalty coefficient transforms constrained optimization problems into unconstrained optimization problems and is used to compare the optimization results with those of the seven other algorithms. The iterative effects of the eight optimization algorithms are shown in Figure 5. A comparison of the results of the welding beam design problem is shown in Table 4.

From the comparison results in Figure 5 and Table 4, it can be deduced that although the SLTSO algorithm may not yield the best individual attribute optimizations for the welded beam, the overall optimization result significantly outperforms the other algorithms. The optimal solution obtained by the SLTSO algorithm for the welding beam is [*x*_1_, *x*_2_, *x*_3_, *x*_4_] = [0.1981, 3.3528, 9.1916, 0.1988], with the optimal objective function value being *f*(*X*) = 1.6713. This indicates that the SLTSO algorithm can deliver the most cost-effective optimal design for the welding beam, thereby validating the SLTSO algorithm’s superior applicability and stability in practical engineering optimization problems.

(2)Gear train design problem

The gear train design problem is an unconstrained discrete design problem in mechanical engineering, which refers to the most simplified gear system [60]. The gear system is defined as the ratio of the output shaft angular velocity to the input shaft angular velocity. The number of teeth *nA*(=*x*_1_), *nB*(=*x*_2_), *nC*(=*x*_3_) and *nD*(=*x*_4_) of the gear are considered as design variables, and their mathematical models are as follows:(43)$$\mathrm{Standard}\;\mathrm{function}:\;f(X)=(\frac{1}{6.931}-\frac{x_{3}x_{2}}{x_{1}x_{4}}{)}^{2}$$
(44)$$\mathrm{Boundary}\;\mathrm{constraints}:\;12\leq x_{i}\leq 60,\;i=1,2,3,4$$
The iterative effect of the gear system design problem using the eight optimization algorithms is shown in Figure 6. A comparison of the results of the gear system design problem is shown in Table 5.

From Figure 6 and Table 5, it can be concluded that in the gear system design benchmark engineering example, the optimal solution of the function optimized by the SLTSO algorithm is [*l*, *t*, *b*, *h*] = [12, 24.0508, 38.6095, 51.8098], and the optimal value is *f*(*X*) = 1.9259 × 10^−32^. Compared with the standard TSO algorithm, the PSO, DBO, SMA, HHO, SABO, and SCSO algorithms have the best computational results and more expressive power, resulting in the best results. The SLTSO algorithm is thus validated to have superior applicability and stability in practical engineering optimization problems for welding beam design and gear system design.

### 6.3. Comparison of UAV Path Planning

A model map of a mountainous environment is an example, where the terrain is characterized by severe undulations and numerous gullies. Environmental disturbances are randomly generated in this terrain, as shown in Figure 7. The starting and ending points of the UAV are set as (10,10,50) and (950950450), respectively (in meters). The number of individuals in the algorithm is uniformly set to 30, and the maximum number of iterations is 50. The SLTSO algorithm and seven other algorithms, including PSO, DBO, SMA, HHO, SABO, SCSO, and standard TSO, were used to solve the model 30 times for each scenario, and the statistical results were obtained. The eight algorithms for path planning and 3D drawing are shown in Figure 8. The top view of path planning for the eight algorithms is shown in Figure 9.

In Figure 8 and Figure 9, it can be seen that the PSO algorithm has the worst quality of the UAV flight route obtained in this flight scenario and the weakest optimization ability, and it is prone to getting stuck in local search, resulting in the worst optimization performance. The TSO algorithm has a stronger evolutionary ability compared with the algorithms included in the comparison, but its local exploration ability needs to be improved. DBO is prone to getting stuck in local search and lacks the ability to jump out of local search, resulting in an uneven flight path. The flight routes obtained by the other algorithms included in the comparison are relatively smooth, but the paths are longer, resulting in higher flight costs. Seven algorithms, including PSO, DBO, SMA, HHO, SABO, SCSO, and standard TSO, have poor performance in solving unmanned aerial vehicle path planning. The obtained unmanned aerial vehicle flight routes are prone to high flight costs and even flight hazards. The search performance of the SABO algorithm is relatively stable, and compared with the other algorithms, it can obtain better UAV flight routes. However, the optimization performance of the algorithm needs to be improved. According to Figure 9, it can be seen that the SLTSO algorithm has a fast convergence speed for solving the UAV path planning problem in this scenario, and the quality of the solution obtained is significantly higher than that of other the algorithms included in the comparison.

The long flight path avoided environmental disturbances, but the low-flying height did not provide safe ground clearance, and the original TSO algorithm iterated into local optima, as shown by comparing the flight routes of TSO and SLTSO. The SLTSO algorithm’s fly route is smooth, eliminating environmental interruptions and providing a safe distance and safe ground clearance from impediments. Figure 8 and Figure 9 show how slowly the original TSO algorithm converges, indicating that it is prone to becoming stuck in local optima. This demonstrates that the SLTSO algorithm can escape local optima. The approach presented in this research, which has a faster convergence time and may quickly leap out of local optimal solutions, based on various performances, is feasible for use in planning high-quality paths in challenging hilly terrain.

## 7. Conclusions

In order to solve the issue of UAV trajectory planning, this paper suggests a sine strategy and a tuna swarm optimization approach based on Levy flight guidance (SLTSO). This method works well for three-dimensional UAV trajectory planning in challenging mission locations. In this algorithm, the population variation in the TSO algorithm is increased by employing the elite opposition-based learning mechanism. The use of a golden sine approach, which effectively synchronizes local excavation ability with global search capacity, accelerates convergence speed. We propose the Levy method to enhance the algorithm’s escape from local optima. A lateral crossover with an adaptive technique is employed to improve the convergence accuracy and global optimization potential of the algorithm. These improved strategies enhance the performance of the SLTSO algorithm in three-dimensional UAV route planning. Experimental data and trajectory planning experiments demonstrate that the SLTSO algorithm can effectively find a safe and effective path. Comparative statistics show that the SLTSO approach outperforms other optimization strategies in terms of addressing UAV route planning difficulties. Regarding UAV route planning, the SLTSO algorithm is a useful and effective method.

Even though the SLTSO approach yields higher-quality UAV route planning solutions, it increases computing complexity. It is necessary to improve the performance of optimization algorithms in the future while preserving a simplified algorithmic mechanism and process. The SLTSO method is expected to find applications in domains such as UAV collaborative route planning and dynamic collision avoidance in the future.

## Figures and Tables

**Figure 1 biomimetics-09-00388-f001:**
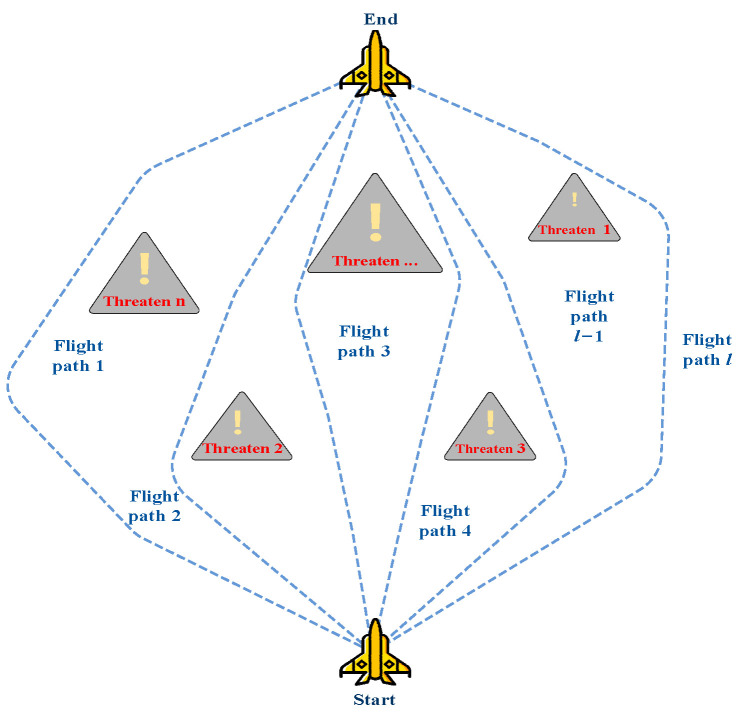
Obstacle avoidance path for UAVs.

**Figure 2 biomimetics-09-00388-f002:**
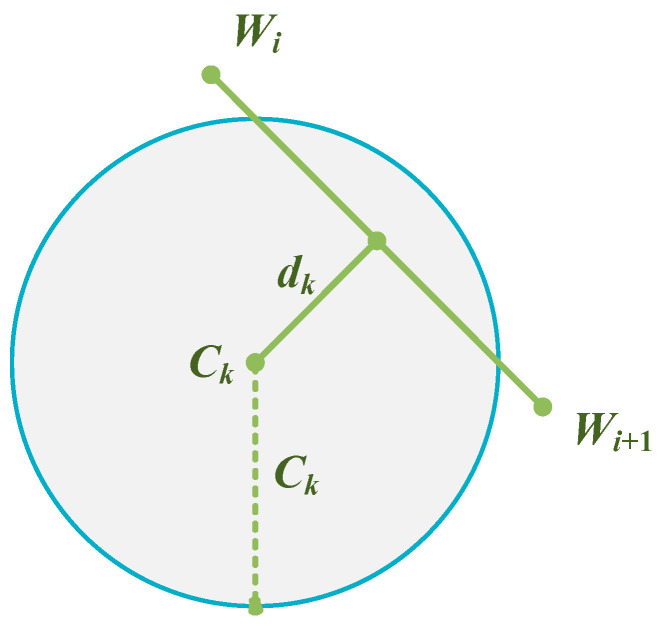
Threat model of the UAV flight.

**Figure 3 biomimetics-09-00388-f003:**
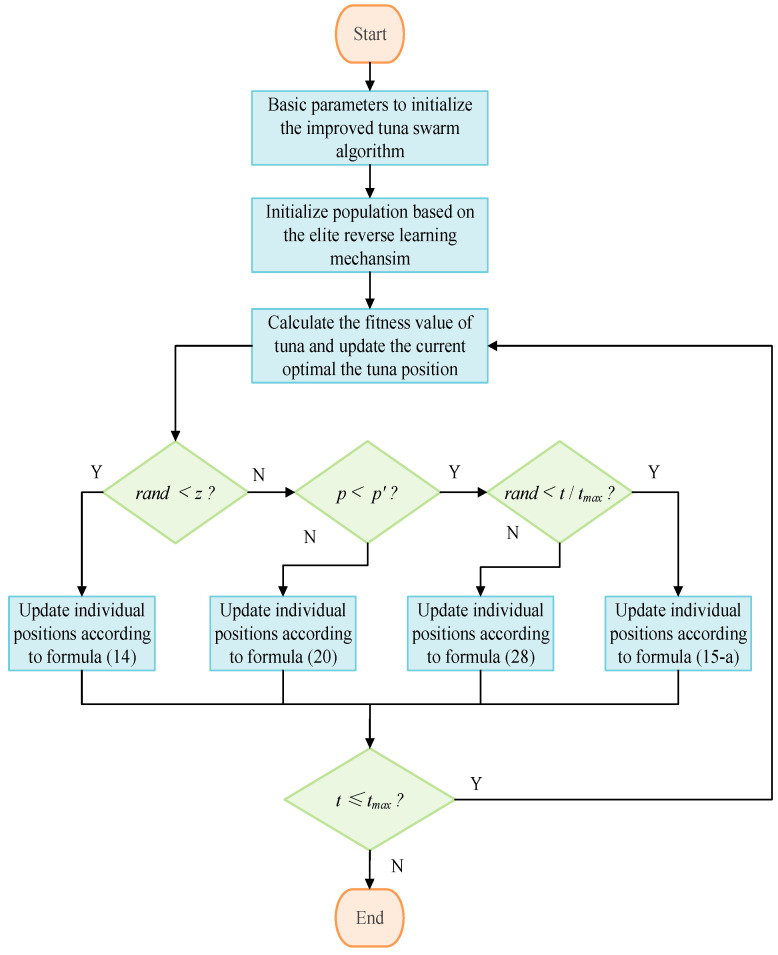
The workflow diagram of the SLTSO algorithm.

**Figure 4 biomimetics-09-00388-f004:**
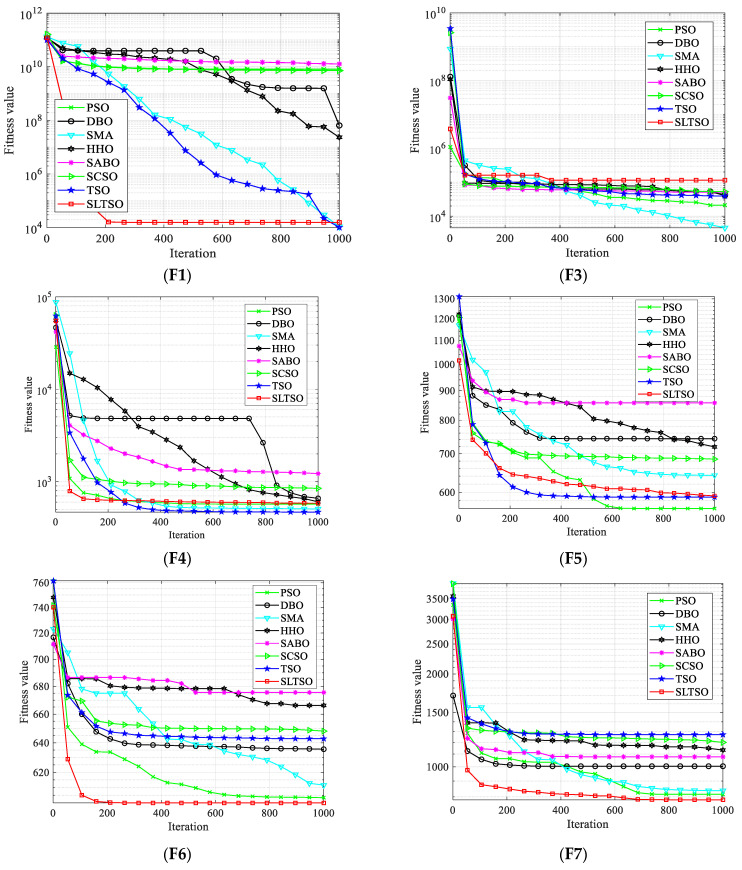
Iterative convergence curves of the test functions.

**Figure 5 biomimetics-09-00388-f005:**
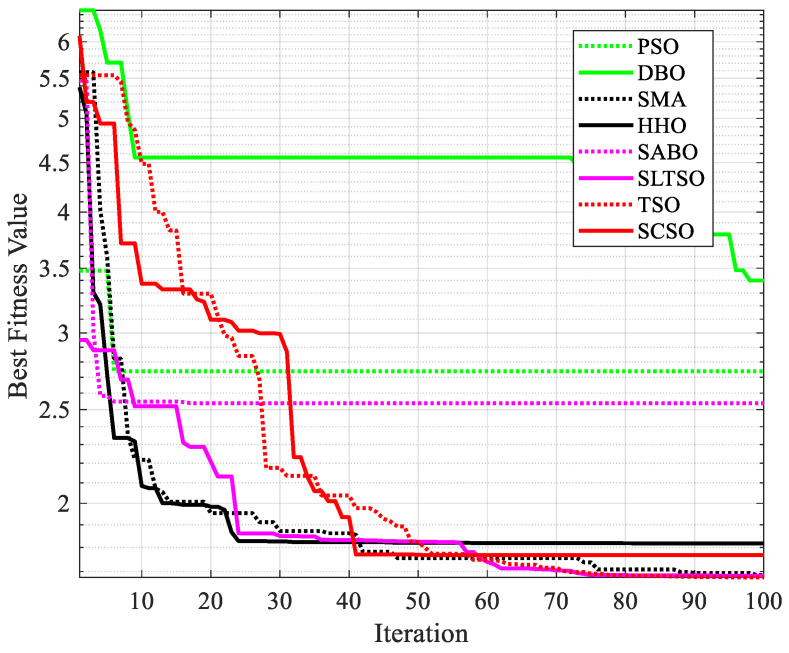
Iterative effect of welding beam design using 8 optimization algorithms.

**Figure 6 biomimetics-09-00388-f006:**
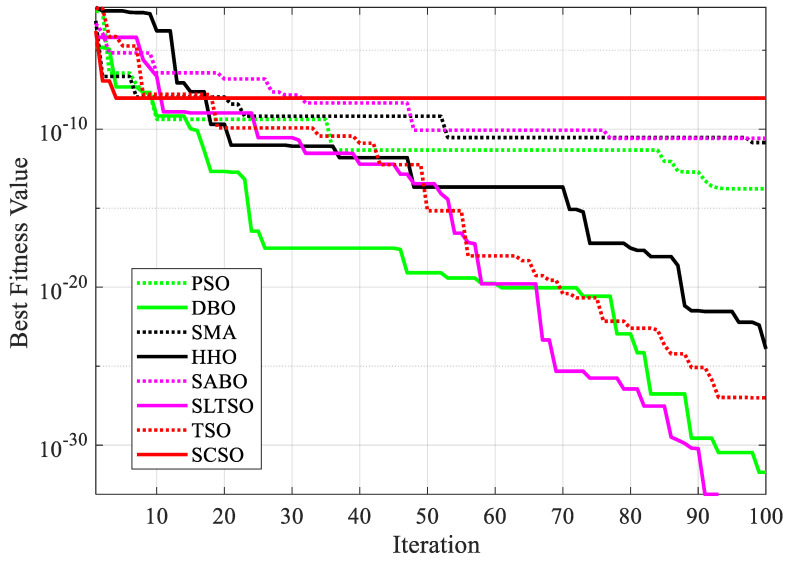
Iterative effect of gear system design using 8 optimization algorithms.

**Figure 7 biomimetics-09-00388-f007:**
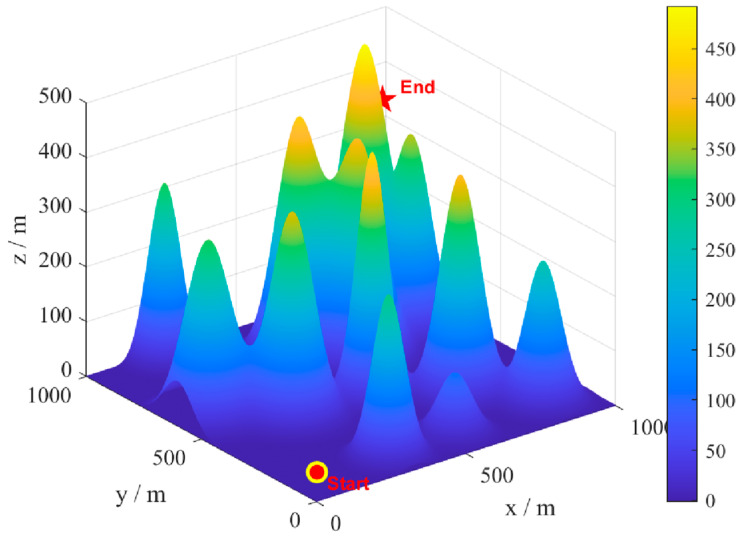
The flight scenario of UAVs.

**Figure 8 biomimetics-09-00388-f008:**
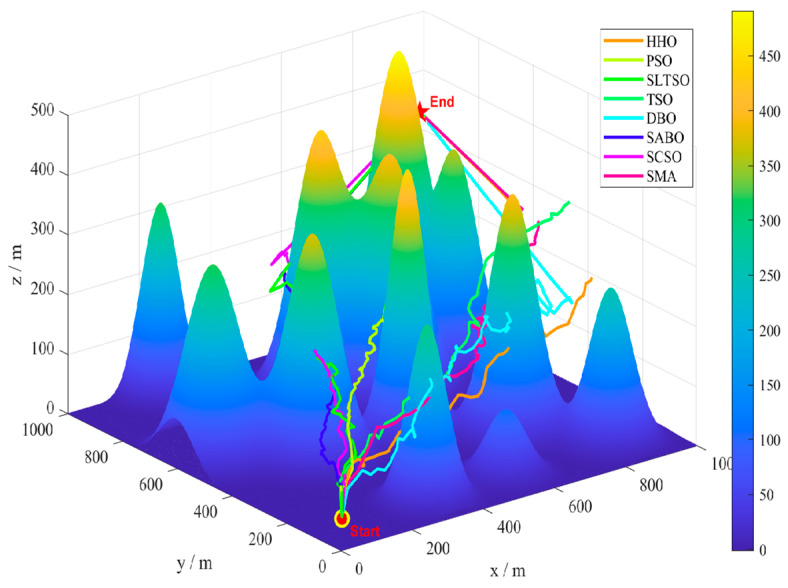
Three-dimensional drawing of path planning for 8 algorithms.

**Figure 9 biomimetics-09-00388-f009:**
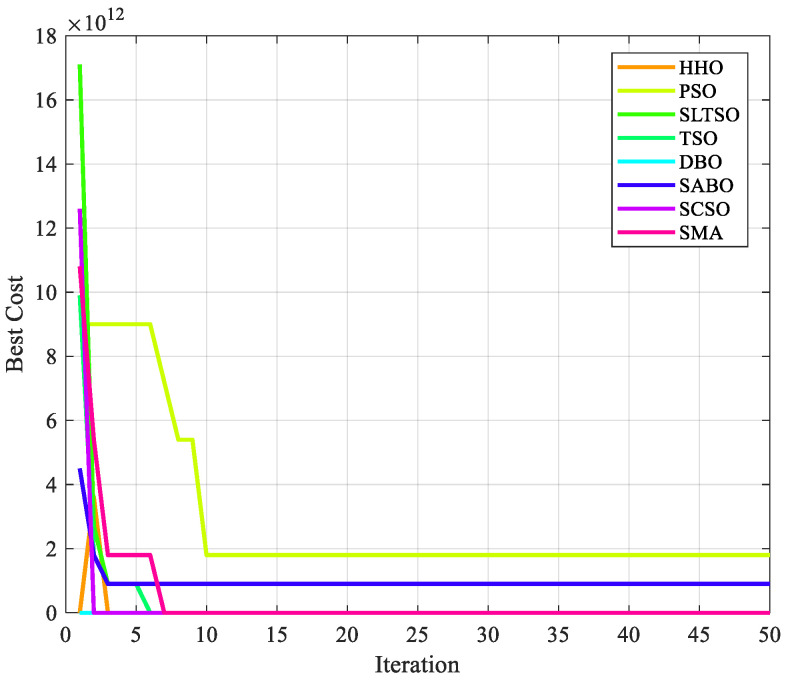
Iterative effect diagram of path planning for 8 algorithms.

**Table 1 biomimetics-09-00388-t001:** Comparison of 8 algorithms of 10 dimension.

Function		PSO	DBO	SMA	HHO	SABO	SCSO	TSO	SLTSO
F1	Min	1.16 × 10^2^	1.02 × 10^2^	9.92 × 10^2^	3.00 × 10^5^	8.98 × 10^6^	2.59 × 10^4^	1.01 × 10^2^	1.05 × 10^2^
	Std	1.28 × 10^7^	1.75 × 10^7^	3.92 × 10^3^	5.84 × 10^5^	6.34 × 10^8^	5.55 × 10^8^	2.57 × 10^3^	2.03 × 10^3^
	Mean	2.34 × 10^6^	5.55 × 10^6^	7.43 × 10^3^	1.11 × 10^6^	4.84 × 10^8^	2.79 × 10^8^	1.59 × 10^3^	2.05 × 10^3^
F3	Min	3.00 × 10^2^	3.00 × 10^2^	3.00 × 10^2^	3.06 × 10^2^	7.68 × 10^2^	3.17 × 10^2^	3.00 × 10^2^	3.00 × 10^2^
	Std	7.73 × 10^−1^	1.55 × 10^3^	6.75 × 10^−1^	3.12 × 10^2^	1.61 × 10^3^	2.49 × 10^3^	1.06 × 10^0^	7.02 × 10^0^
	Mean	3.00 × 10^2^	1.37 × 10^3^	3.00 × 10^2^	6.16 × 10^2^	3.09 × 10^3^	2.87 × 10^3^	3.00 × 10^2^	3.02 × 10^2^
F4	Min	4.01 × 10^2^	4.01 × 10^2^	4.00 × 10^2^	4.00 × 10^2^	4.07 × 10^2^	4.03 × 10^2^	4.00 × 10^2^	4.00 × 10^2^
	Std	4.21 × 10^1^	3.28 × 10^1^	3.21 × 10^1^	3.54 × 10^1^	2.22 × 10^1^	3.89 × 10^1^	1.30 × 10^1^	2.01 × 10^1^
	Mean	4.23 × 10^2^	4.23 × 10^2^	4.22 × 10^2^	4.27 × 10^2^	4.43 × 10^2^	4.41 × 10^2^	4.06 × 10^2^	4.11 × 10^2^
F5	Min	5.05 × 10^2^	5.17 × 10^2^	5.09 × 10^2^	5.24 × 10^2^	5.34 × 10^2^	5.12 × 10^2^	5.12 × 10^2^	5.02 × 10^2^
	Std	8.53 × 100	1.37 × 10^1^	6.52 × 10^0^	1.72 × 10^1^	1.03 × 10^1^	1.55 × 10^1^	1.24 × 10^1^	2.37 × 10^0^
	Mean	5.14 × 10^2^	5.40 × 10^2^	5.20 × 10^2^	5.59 × 10^2^	5.54 × 10^2^	5.41 × 10^2^	5.27 × 10^2^	5.07 × 10^2^
F6	Min	6.00 × 10^2^	6.01 × 10^2^	6.00 × 10^2^	6.14 × 10^2^	6.09 × 10^2^	6.03 × 10^2^	6.02 × 10^2^	6.00 × 10^2^
	Std	2.89 × 10^−1^	8.23 × 10^0^	6.09 × 10^−1^	1.25 × 10^1^	1.03 × 10^1^	1.07 × 10^1^	6.83 × 10^0^	9.05 × 10^−9^
	Mean	6.00 × 10^2^	6.13 × 10^2^	6.00 × 10^2^	6.42 × 10^2^	6.20 × 10^2^	6.18 × 10^2^	6.11 × 10^2^	6.00 × 10^2^
F7	Min	7.13 × 10^2^	7.30 × 10^2^	7.19 × 10^2^	7.62 × 10^2^	7.39 × 10^2^	7.39 × 10^2^	7.22 × 10^2^	7.06 × 10^2^
	Std	6.44 × 100	1.45 × 10^1^	5.96 × 10^0^	1.83 × 10^1^	1.39 × 10^1^	1.68 × 10^1^	1.57 × 10^1^	3.46 × 10^0^
	Mean	7.23 × 10^2^	7.49 × 10^2^	7.29 × 10^2^	7.91 × 10^2^	7.67 × 10^2^	7.68 × 10^2^	7.48 × 10^2^	7.16 × 10^2^
F8	Min	8.05 × 10^2^	8.07 × 10^2^	8.05 × 10^2^	8.15 × 10^2^	8.32 × 10^2^	8.18 × 10^2^	8.06 × 10^2^	8.02 × 10^2^
	Std	4.71 × 100	1.58 × 10^1^	8.80 × 10^0^	8.53 × 10^0^	9.17 × 10^0^	7.51 × 10^0^	7.37 × 10^0^	2.10 × 10^0^
	Mean	8.13 × 10^2^	8.37 × 10^2^	8.20 × 10^2^	8.30 × 10^2^	8.49 × 10^2^	8.31 × 10^2^	8.22 × 10^2^	8.07 × 10^2^
F9	Min	9.00 × 10^2^	9.02 × 10^2^	9.00 × 10^2^	9.82 × 10^2^	9.18 × 10^2^	9.15 × 10^2^	9.11 × 10^2^	9.00 × 10^2^
	Std	1.16 × 10^−1^	1.14 × 10^2^	2.94 × 10^0^	2.14 × 10^2^	9.53 × 10^1^	2.33 × 10^2^	7.82 × 10^1^	8.29 × 10^−2^
	Mean	9.00 × 10^2^	9.98 × 10^2^	9.01 × 10^2^	1.58 × 10^3^	1.04 × 10^3^	1.12 × 10^3^	9.96 × 10^2^	9.00 × 10^2^
F10	Min	1.02 × 10^3^	1.28 × 10^3^	1.14 × 10^3^	1.27 × 10^3^	2.34 × 10^3^	1.37 × 10^3^	1.37 × 10^3^	1.05 × 10^3^
	Std	2.55 × 10^2^	3.10 × 10^2^	2.52 × 10^2^	3.65 × 10^2^	1.98 × 10^2^	3.20 × 10^2^	3.00 × 10^2^	1.71 × 10^2^
	Mean	1.58 × 10^3^	1.98 × 10^3^	1.59 × 10^3^	2.03 × 10^3^	2.81 × 10^3^	2.08 × 10^3^	1.96 × 10^3^	1.49 × 10^3^
F11	Min	1.10 × 10^3^	1.11 × 10^3^	1.11 × 10^3^	1.11 × 10^3^	1.14 × 10^3^	1.11 × 10^3^	1.11 × 10^3^	1.10 × 10^3^
	Std	2.97 × 10^1^	1.22 × 10^2^	1.04 × 10^2^	5.38 × 10^1^	1.37 × 10^3^	4.90 × 10^1^	2.38 × 10^1^	3.01 × 10^0^
	Mean	1.12 × 10^3^	1.23 × 10^3^	1.21 × 10^3^	1.18 × 10^3^	1.86 × 10^3^	1.16 × 10^3^	1.14 × 10^3^	1.10 × 10^3^
F12	Min	3.65 × 10^3^	1.93 × 10^3^	6.40 × 10^3^	5.05 × 10^4^	1.41 × 10^5^	1.79 × 10^4^	2.27 × 10^3^	1.98 × 10^3^
	Std	1.49 × 10^6^	5.67 × 10^6^	6.24 × 10^5^	2.98 × 10^6^	1.72 × 10^6^	1.95 × 10^6^	9.44 × 10^3^	1.20 × 10^4^
	Mean	2.96 × 10^5^	3.41 × 10^6^	4.68 × 10^5^	2.46 × 10^6^	2.01 × 10^6^	1.78 × 10^6^	9.76 × 10^3^	1.35 × 10^4^
F13	Min	1.38 × 10^3^	1.51 × 10^3^	1.55 × 10^3^	2.45 × 10^3^	3.56 × 10^3^	2.47 × 10^3^	1.58 × 10^3^	1.36 × 10^3^
	Std	9.29 × 10^3^	1.19 × 10^4^	1.24 × 10^4^	8.47 × 10^3^	6.16 × 10^3^	1.05 × 10^4^	6.04 × 10^3^	1.32 × 10^4^
	Mean	9.63 × 10^3^	1.45 × 10^4^	1.12 × 10^4^	1.42 × 10^4^	1.30 × 10^4^	1.43 × 10^4^	1.00 × 10^4^	1.24 × 10^4^
F14	Min	1.43 × 10^3^	1.45 × 10^3^	1.43 × 10^3^	1.50 × 10^3^	1.51 × 10^3^	1.46 × 10^3^	1.43 × 10^3^	1.41 × 10^3^
	Std	8.39 × 10^2^	5.29 × 10^2^	2.57 × 10^3^	9.72 × 10^2^	1.05 × 10^3^	2.03 × 10^3^	3.96 × 10^1^	2.49 × 10^2^
	Mean	1.76 × 10^3^	1.97 × 10^3^	2.16 × 10^3^	2.09 × 10^3^	2.21 × 10^3^	3.48 × 10^3^	1.50 × 10^3^	1.56 × 10^3^
F15	Min	1.51 × 10^3^	1.63 × 10^3^	1.53 × 10^3^	1.78 × 10^3^	2.75 × 10^3^	1.58 × 10^3^	1.52 × 10^3^	1.51 × 10^3^
	Std	2.35 × 10^3^	2.45 × 10^3^	3.62 × 10^3^	3.00 × 10^3^	5.27 × 10^3^	1.80 × 10^3^	3.86 × 10^2^	7.64 × 10^2^
	Mean	3.10 × 10^3^	4.34 × 10^3^	5.19 × 10^3^	7.92 × 10^3^	9.55 × 10^3^	4.92 × 10^3^	1.81 × 10^3^	1.86 × 10^3^
F16	Min	1.60 × 10^3^	1.61 × 10^3^	1.60 × 10^3^	1.63 × 10^3^	1.79 × 10^3^	1.62 × 10^3^	1.60 × 10^3^	1.60 × 10^3^
	Std	1.37 × 10^2^	1.45 × 10^2^	9.73 × 10^1^	1.49 × 10^2^	1.23 × 10^2^	1.34 × 10^2^	1.19 × 10^2^	4.60 × 10^1^
	Mean	1.76 × 10^3^	1.82 × 10^3^	1.70 × 10^3^	1.93 × 10^3^	2.06 × 10^3^	1.82 × 10^3^	1.79 × 10^3^	1.63 × 10^3^
F17	Min	1.71 × 10^3^	1.73 × 10^3^	1.72 × 10^3^	1.75 × 10^3^	1.76 × 10^3^	1.73 × 10^3^	1.72 × 10^3^	1.70 × 10^3^
	Std	3.24 × 10^1^	4.04 × 10^1^	3.63 × 10^1^	5.87 × 10^1^	7.79 × 10^1^	2.98 × 10^1^	2.32 × 10^1^	2.49 × 10^1^
	Mean	1.75 × 10^3^	1.78 × 10^3^	1.76 × 10^3^	1.80 × 10^3^	1.87 × 10^3^	1.77 × 10^3^	1.75 × 10^3^	1.72 × 10^3^
F18	Min	2.23 × 10^3^	2.18 × 10^3^	4.46 × 10^3^	2.89 × 10^3^	2.55 × 10^3^	4.08 × 10^3^	2.01 × 10^3^	1.97 × 10^3^
	Std	1.52 × 10^4^	1.62 × 10^4^	1.40 × 10^4^	1.33 × 10^4^	1.21 × 10^4^	1.77 × 10^4^	7.14 × 10^3^	7.76 × 10^3^
	Mean	2.27 × 10^4^	1.91 × 10^4^	2.57 × 10^4^	1.89 × 10^4^	1.88 × 10^4^	2.41 × 10^4^	9.01 × 10^3^	8.51 × 10^3^
F19	Min	1.96 × 10^3^	1.97 × 10^3^	1.93 × 10^3^	1.94 × 10^3^	3.05 × 10^3^	1.95 × 10^3^	1.94 × 10^3^	1.91 × 10^3^
	Std	6.92 × 10^3^	2.06 × 10^4^	1.05 × 10^4^	1.87 × 10^4^	5.33 × 10^3^	6.89 × 10^3^	7.06 × 10^2^	9.75 × 10^2^
	Mean	6.29 × 10^3^	1.14 × 10^4^	1.49 × 10^4^	2.08 × 10^4^	8.90 × 10^3^	8.01 × 10^3^	2.56 × 10^3^	2.57 × 10^3^
F20	Min	2.00 × 10^3^	2.03 × 10^3^	2.00 × 10^3^	2.04 × 10^3^	2.12 × 10^3^	2.04 × 10^3^	2.01 × 10^3^	2.00 × 10^3^
	Std	5.44 × 10^1^	7.86 × 10^1^	5.01 × 10^1^	6.89 × 10^1^	6.66 × 10^1^	6.78 × 10^1^	6.41 × 10^1^	2.23 × 10^1^
	Mean	2.05 × 10^3^	2.12 × 10^3^	2.05 × 10^3^	2.18 × 10^3^	2.23 × 10^3^	2.14 × 10^3^	2.10 × 10^3^	2.01 × 10^3^
F21	Min	2.20 × 10^3^	2.20 × 10^3^	2.10 × 10^3^	2.20 × 10^3^	2.33 × 10^3^	2.20 × 10^3^	2.20 × 10^3^	2.20 × 10^3^
	Std	4.09 × 10^1^	5.52 × 10^1^	5.58 × 10^1^	7.04 × 10^1^	1.52 × 10^1^	5.48 × 10^1^	4.02 × 10^1^	5.17 × 10^1^
	Mean	2.30 × 10^3^	2.23 × 10^3^	2.31 × 10^3^	2.32 × 10^3^	2.35 × 10^3^	2.31 × 10^3^	2.22 × 10^3^	2.27 × 10^3^
F22	Min	2.23 × 10^3^	2.24 × 10^3^	2.22 × 10^3^	2.26 × 10^3^	2.31 × 10^3^	2.30 × 10^3^	2.24 × 10^3^	2.20 × 10^3^
	Std	2.93 × 10^1^	2.31 × 10^1^	2.72 × 10^2^	2.10 × 10^2^	5.19 × 10^1^	7.85 × 10^1^	1.60 × 10^1^	1.95 × 10^1^
	Mean	2.31 × 10^3^	2.31 × 10^3^	2.41 × 10^3^	2.35 × 10^3^	2.34 × 10^3^	2.34 × 10^3^	2.30 × 10^3^	2.30 × 10^3^
F23	Min	2.61 × 10^3^	2.62 × 10^3^	2.61 × 10^3^	2.62 × 10^3^	2.63 × 10^3^	2.61 × 10^3^	2.61 × 10^3^	2.61 × 10^3^
	Std	1.22 × 10^1^	1.78 × 10^1^	7.85 × 10^0^	2.99 × 10^1^	2.01 × 10^1^	1.52 × 10^1^	1.42 × 10^1^	2.89 × 10^0^
	Mean	2.63 × 10^3^	2.64 × 10^3^	2.62 × 10^3^	2.68 × 10^3^	2.66 × 10^3^	2.64 × 10^3^	2.63 × 10^3^	2.61 × 10^3^
F24	Min	2.50 × 10^3^	2.50 × 10^3^	2.74 × 10^3^	2.50 × 10^3^	2.53 × 10^3^	2.50 × 10^3^	2.50 × 10^3^	2.50 × 10^3^
	Std	4.93 × 10^1^	1.06 × 10^2^	1.26 × 10^1^	1.43 × 10^2^	4.98 × 10^1^	7.23 × 10^1^	8.02 × 10^1^	9.11 × 10^1^
	Mean	2.76 × 10^3^	2.72 × 10^3^	2.76 × 10^3^	2.78 × 10^3^	2.78 × 10^3^	2.75 × 10^3^	2.73 × 10^3^	2.70 × 10^3^
F25	Min	2.90 × 10^3^	2.90 × 10^3^	2.90 × 10^3^	2.90 × 10^3^	2.94 × 10^3^	2.91 × 10^3^	2.90 × 10^3^	2.90 × 10^3^
	Std	3.18 × 10^1^	2.48 × 10^1^	3.78 × 10^1^	2.17 × 10^1^	2.25 × 10^1^	3.08 × 10^1^	2.42 × 10^1^	2.33 × 10^1^
	Mean	2.93 × 10^3^	2.94 × 10^3^	2.94 × 10^3^	2.94 × 10^3^	2.97 × 10^3^	2.94 × 10^3^	2.93 × 10^3^	2.92 × 10^3^
F26	Min	2.60 × 10^3^	2.82 × 10^3^	2.90 × 10^3^	2.61 × 10^3^	3.11 × 10^3^	2.86 × 10^3^	2.60 × 10^3^	2.60 × 10^3^
	Std	3.36 × 10^2^	2.31 × 10^2^	4.42 × 10^2^	6.56 × 10^2^	4.02 × 10^2^	2.28 × 10^2^	1.43 × 10^2^	1.29 × 10^2^
	Mean	3.08 × 10^3^	3.17 × 10^3^	3.19 × 10^3^	3.56 × 10^3^	3.47 × 10^3^	3.17 × 10^3^	2.96 × 10^3^	2.92 × 10^3^
F27	Min	3.09 × 10^3^	3.09 × 10^3^	3.09 × 10^3^	3.11 × 10^3^	3.10 × 10^3^	3.09 × 10^3^	3.09 × 10^3^	3.07 × 10^3^
	Std	2.52 × 10^1^	1.02 × 10^1^	1.91 × 10^0^	4.97 × 10^1^	1.84 × 10^1^	2.84 × 10^1^	2.55 × 10^1^	2.26 × 10^0^
	Mean	3.12 × 10^3^	3.10 × 10^3^	3.09 × 10^3^	3.17 × 10^3^	3.11 × 10^3^	3.11 × 10^3^	3.11 × 10^3^	3.07 × 10^3^
F28	Min	3.10 × 10^3^	3.17 × 10^3^	3.17 × 10^3^	3.10 × 10^3^	3.23 × 10^3^	3.17 × 10^3^	3.10 × 10^3^	3.10 × 10^3^
	Std	1.22 × 10^2^	1.27 × 10^2^	1.30 × 10^2^	1.12 × 10^2^	1.38 × 10^2^	1.18 × 10^2^	1.61 × 10^2^	4.61 × 10^1^
	Mean	3.39 × 10^3^	3.35 × 10^3^	3.41 × 10^3^	3.38 × 10^3^	3.50 × 10^3^	3.32 × 10^3^	3.40 × 10^3^	3.26 × 10^3^
F29	Min	3.14 × 10^3^	3.16 × 10^3^	3.13 × 10^3^	3.21 × 10^3^	3.20 × 10^3^	3.17 × 10^3^	3.16 × 10^3^	3.15 × 10^3^
	Std	4.47 × 10^1^	8.95 × 10^1^	5.93 × 10^1^	8.34 × 10^1^	9.03 × 10^1^	8.41 × 10^1^	6.59 × 10^1^	1.27 × 10^1^
	Mean	3.20 × 10^3^	3.30 × 10^3^	3.22 × 10^3^	3.36 × 10^3^	3.34 × 10^3^	3.29 × 10^3^	3.25 × 10^3^	3.17 × 10^3^
F30	Min	6.30 × 10^3^	6.66 × 10^3^	6.71 × 10^3^	1.68 × 10^4^	5.59 × 10^4^	5.04 × 10^3^	4.67 × 10^3^	3.23 × 10^3^
	Std	5.61 × 10^5^	1.44 × 10^6^	6.02 × 10^5^	2.77 × 10^6^	4.55 × 10^6^	1.31 × 10^6^	2.12 × 10^6^	1.25 × 10^3^
	Mean	4.25 × 10^5^	1.21 × 10^6^	3.66 × 10^5^	2.07 × 10^6^	2.52 × 10^6^	1.11 × 10^6^	7.61 × 10^5^	3.71 × 10^3^

**Table 2 biomimetics-09-00388-t002:** Comparison of 8 algorithms of 30 dimension.

Function		PSO	DBO	SMA	HHO	SABO	SCSO	TSO	SLTSO
F1	Min	2.18 × 10^3^	5.32 × 10^7^	4.58 × 10^4^	8.83 × 10^7^	3.71 × 10^9^	8.02 × 10^8^	1.11 × 10^6^	1.06 × 10^2^
	Std	3.09 × 10^9^	1.77 × 10^8^	4.53 × 10^4^	2.43 × 10^8^	5.66 × 10^9^	3.97 × 10^9^	1.49 × 10^7^	9.81 × 10^3^
	Mean	2.00 × 10^9^	2.59 × 10^8^	1.05 × 10^5^	4.30 × 10^8^	1.34 × 10^10^	9.08 × 10^9^	1.07 × 10^7^	7.75 × 10^3^
F3	Min	3.99 × 10^4^	6.29 × 10^4^	1.39 × 10^4^	3.80 × 10^4^	4.59 × 10^4^	3.58 × 10^4^	2.48 × 10^4^	7.14 × 10^4^
	Std	2.50 × 10^4^	9.19 × 10^3^	1.55 × 10^4^	8.27 × 10^3^	1.02 × 10^4^	1.10 × 10^4^	1.51 × 10^4^	3.48 × 10^4^
	Mean	7.43 × 10^4^	8.57 × 10^4^	3.52 × 10^4^	5.76 × 10^4^	6.57 × 10^4^	6.07 × 10^4^	6.08 × 10^4^	1.43 × 10^5^
F4	Min	4.92 × 10^2^	5.34 × 10^2^	4.76 × 10^2^	5.96 × 10^2^	9.13 × 10^2^	5.31 × 10^2^	4.86 × 10^2^	4.18 × 10^2^
	Std	3.86 × 10^2^	1.63 × 10^2^	1.50 × 10^1^	9.82 × 10^1^	1.10 × 10^3^	4.74 × 10^2^	2.51 × 10^1^	3.41 × 10^1^
	Mean	7.68 × 10^2^	6.68 × 10^2^	5.02 × 10^2^	7.29 × 10^2^	2.21 × 10^3^	9.90 × 10^2^	5.31 × 10^2^	4.73 × 10^2^
F5	Min	5.40 × 10^2^	6.68 × 10^2^	5.61 × 10^2^	7.24 × 10^2^	7.64 × 10^2^	6.84 × 10^2^	6.00 × 10^2^	5.60 × 10^2^
	Std	2.33 × 10^1^	5.76 × 10^1^	4.59 × 10^1^	2.64 × 10^1^	3.60 × 10^1^	4.98 × 10^1^	3.02 × 10^1^	1.75 × 10^1^
	Mean	5.83 × 10^2^	7.59 × 10^2^	6.37 × 10^2^	7.70 × 10^2^	8.27 × 10^2^	7.67 × 10^2^	6.65 × 10^2^	5.94 × 10^2^
F6	Min	6.02 × 10^2^	6.18 × 10^2^	6.07 × 10^2^	6.58 × 10^2^	6.37 × 10^2^	6.40 × 10^2^	6.25 × 10^2^	6.00 × 10^2^
	Std	6.16 × 10^0^	1.09 × 10^1^	1.02 × 10^1^	5.11 × 10^0^	1.47 × 10^1^	1.08 × 10^1^	7.91 × 10^0^	1.19 × 10^−2^
	Mean	6.08 × 10^2^	6.46 × 10^2^	6.18 × 10^2^	6.69 × 10^2^	6.67 × 10^2^	6.62 × 10^2^	6.44 × 10^2^	6.00 × 10^2^
F7	Min	7.93 × 10^2^	8.54 × 10^2^	8.32 × 10^2^	1.18 × 10^3^	1.04 × 10^3^	9.62 × 10^2^	9.61 × 10^2^	8.03 × 10^2^
	Std	4.86 × 10^1^	9.67 × 10^1^	3.99 × 10^1^	6.13 × 10^1^	5.97 × 10^1^	1.07 × 10^2^	7.92 × 10^1^	1.98 × 10^1^
	Mean	8.44 × 10^2^	1.02 × 10^3^	8.93 × 10^2^	1.30 × 10^3^	1.13 × 10^3^	1.16 × 10^3^	1.08 × 10^3^	8.36 × 10^2^
F8	Min	8.44 × 10^2^	9.21 × 10^2^	8.73 × 10^2^	9.26 × 10^2^	1.04 × 10^3^	9.56 × 10^2^	8.67 × 10^2^	8.56 × 10^2^
	Std	2.69 × 10^1^	5.11 × 10^1^	2.86 × 10^1^	3.00 × 10^1^	2.60 × 10^1^	2.88 × 10^1^	3.19 × 10^1^	1.69 × 10^1^
	Mean	8.90 × 10^2^	1.02 × 10^3^	9.19 × 10^2^	9.92 × 10^2^	1.08 × 10^3^	1.01 × 10^3^	9.30 × 10^2^	8.83 × 10^2^
F9	Min	9.24 × 10^2^	3.52 × 10^3^	2.16 × 10^3^	6.48 × 10^3^	3.84 × 10^3^	3.65 × 10^3^	2.46 × 10^3^	9.01 × 10^2^
	Std	7.47 × 10^2^	1.86 × 10^3^	1.65 × 10^3^	1.38 × 10^3^	1.74 × 10^3^	1.58 × 10^3^	1.36 × 10^3^	3.57 × 10^2^
	Mean	1.55 × 10^3^	6.49 × 10^3^	4.75 × 10^3^	8.78 × 10^3^	6.71 × 10^3^	6.47 × 10^3^	4.58 × 10^3^	1.08 × 10^3^
F10	Min	2.71 × 10^3^	4.41 × 10^3^	3.98 × 10^3^	5.31 × 10^3^	7.90 × 10^3^	4.80 × 10^3^	4.12 × 10^3^	3.89 × 10^3^
	Std	7.57 × 10^2^	1.14 × 10^3^	5.90 × 10^2^	7.52 × 10^2^	3.81 × 10^2^	6.72 × 10^2^	1.14 × 10^3^	4.80 × 10^2^
	Mean	4.58 × 10^3^	6.84 × 10^3^	4.94 × 10^3^	6.39 × 10^3^	8.77 × 10^3^	6.10 × 10^3^	6.09 × 10^3^	5.12 × 10^3^
F11	Min	1.16 × 10^3^	1.29 × 10^3^	1.18 × 10^3^	1.28 × 10^3^	2.31 × 10^3^	1.52 × 10^3^	1.18 × 10^3^	1.15 × 10^3^
	Std	1.17 × 10^2^	1.10 × 10^3^	7.79 × 10^1^	2.41 × 10^2^	2.16 × 10^3^	1.46 × 10^3^	4.32 × 10^1^	2.73 × 10^1^
	Mean	1.31 × 10^3^	2.10 × 10^3^	1.29 × 10^3^	1.59 × 10^3^	5.60 × 10^3^	3.25 × 10^3^	1.26 × 10^3^	1.18 × 10^3^
F12	Min	1.85 × 10^5^	1.28 × 10^6^	3.26 × 10^5^	5.54 × 10^6^	1.09 × 10^8^	2.43 × 10^7^	4.06 × 10^5^	4.83 × 10^4^
	Std	1.46 × 10^8^	1.40 × 10^8^	3.58 × 10^6^	7.73 × 10^7^	7.16 × 10^8^	7.65 × 10^8^	2.85 × 10^6^	1.81 × 10^6^
	Mean	7.63 × 10^7^	9.69 × 10^7^	4.34 × 10^6^	6.83 × 10^7^	9.74 × 10^8^	4.87 × 10^8^	3.16 × 10^6^	1.79 × 10^6^
F13	Min	7.28 × 10^3^	1.10 × 10^4^	1.46 × 10^4^	3.30 × 10^5^	2.55 × 10^6^	2.46 × 10^4^	4.30 × 10^3^	1.36 × 10^3^
	Std	2.68 × 10^8^	1.73 × 10^7^	5.33 × 10^4^	7.42 × 10^5^	2.92 × 10^8^	8.98 × 10^7^	1.29 × 10^4^	1.73 × 10^4^
	Mean	7.66 × 10^7^	6.67 × 10^6^	6.64 × 10^4^	1.04 × 10^6^	1.50 × 10^8^	3.83 × 10^7^	1.93 × 10^4^	1.86 × 10^4^
F14	Min	5.00 × 10^3^	1.32 × 10^4^	1.43 × 10^4^	2.16 × 10^4^	4.68 × 10^4^	1.57 × 10^4^	3.38 × 10^3^	3.82 × 10^3^
	Std	9.64 × 10^4^	4.69 × 10^5^	2.88 × 10^5^	1.07 × 10^6^	8.95 × 10^5^	8.53 × 10^5^	4.70 × 10^4^	6.49 × 10^4^
	Mean	1.04 × 10^5^	3.84 × 10^5^	2.84 × 10^5^	1.15 × 10^6^	1.17 × 10^6^	6.35 × 10^5^	3.83 × 10^4^	6.54 × 10^4^
F15	Min	2.20 × 10^3^	9.46 × 10^3^	3.45 × 10^3^	3.16 × 10^4^	1.45 × 10^5^	2.43 × 10^4^	1.80 × 10^3^	1.66 × 10^3^
	Std	2.49 × 10^4^	2.35 × 10^6^	1.62 × 10^4^	1.08 × 10^5^	5.09 × 10^6^	2.58 × 10^6^	1.16 × 10^4^	1.12 × 10^4^
	Mean	2.11 × 10^4^	5.86 × 10^5^	2.15 × 10^4^	1.40 × 10^5^	3.13 × 10^6^	2.06 × 10^6^	7.75 × 10^3^	1.22 × 10^4^
F16	Min	1.95 × 10^3^	2.59 × 10^3^	2.00 × 10^3^	2.28 × 10^3^	3.75 × 10^3^	2.43 × 10^3^	2.37 × 10^3^	2.14 × 10^3^
	Std	2.99 × 10^2^	4.02 × 10^2^	3.34 × 10^2^	5.15 × 10^2^	2.86 × 10^2^	3.67 × 10^2^	2.95 × 10^2^	2.36 × 10^2^
	Mean	2.60 × 10^3^	3.25 × 10^3^	2.67 × 10^3^	3.71 × 10^3^	4.17 × 10^3^	3.30 × 10^3^	2.85 × 10^3^	2.60 × 10^3^
F17	Min	1.79 × 10^3^	1.92 × 10^3^	1.91 × 10^3^	1.81 × 10^3^	2.44 × 10^3^	1.89 × 10^3^	1.97 × 10^3^	1.79 × 10^3^
	Std	1.67 × 10^2^	3.43 × 10^2^	2.64 × 10^2^	3.16 × 10^2^	2.42 × 10^2^	2.48 × 10^2^	2.11 × 10^2^	1.78 × 10^2^
	Mean	2.11 × 10^3^	2.68 × 10^3^	2.32 × 10^3^	2.71 × 10^3^	2.92 × 10^3^	2.38 × 10^3^	2.38 × 10^3^	2.13 × 10^3^
F18	Min	1.70 × 10^5^	7.87 × 10^4^	3.53 × 10^5^	1.83 × 10^5^	2.69 × 10^5^	5.12 × 10^4^	4.72 × 10^4^	1.83 × 10^5^
	Std	1.14 × 10^6^	4.03 × 10^6^	2.78 × 10^6^	4.47 × 10^6^	6.12 × 10^6^	7.48 × 10^6^	2.06 × 10^5^	6.45 × 10^5^
	Mean	1.31 × 10^6^	3.11 × 10^6^	2.99 × 10^6^	4.09 × 10^6^	4.56 × 10^6^	4.71 × 10^6^	2.93 × 10^5^	1.06 × 10^6^
F19	Min	1.98 × 10^3^	2.93 × 10^3^	2.14 × 10^3^	1.18 × 10^5^	2.75 × 10^5^	1.55 × 10^5^	2.05 × 10^3^	1.99 × 10^3^
	Std	1.10 × 10^5^	6.97 × 10^7^	1.99 × 10^4^	1.37 × 10^6^	4.73 × 10^6^	1.59 × 10^6^	9.15 × 10^3^	1.44 × 10^4^
	Mean	5.21 × 10^4^	1.69 × 10^7^	1.98 × 10^4^	1.75 × 10^6^	5.32 × 10^6^	1.89 × 10^6^	1.05 × 10^4^	1.32 × 10^4^
F20	Min	2.07 × 10^3^	2.30 × 10^3^	2.09 × 10^3^	2.54 × 10^3^	2.80 × 10^3^	2.30 × 10^3^	2.23 × 10^3^	2.16 × 10^3^
	Std	1.75 × 10^2^	2.38 × 10^2^	2.20 × 10^2^	1.96 × 10^2^	1.50 × 10^2^	2.02 × 10^2^	2.01 × 10^2^	1.25 × 10^2^
	Mean	2.42 × 10^3^	2.71 × 10^3^	2.54 × 10^3^	2.84 × 10^3^	3.08 × 10^3^	2.69 × 10^3^	2.58 × 10^3^	2.37 × 10^3^
F21	Min	2.35 × 10^3^	2.42 × 10^3^	2.38 × 10^3^	2.47 × 10^3^	2.54 × 10^3^	2.46 × 10^3^	2.39 × 10^3^	2.37 × 10^3^
	Std	3.43 × 10^1^	5.40 × 10^1^	2.56 × 10^1^	5.86 × 10^1^	3.77 × 10^1^	4.58 × 10^1^	3.06 × 10^1^	1.20 × 10^1^
	Mean	2.41 × 10^3^	2.55 × 10^3^	2.43 × 10^3^	2.60 × 10^3^	2.60 × 10^3^	2.53 × 10^3^	2.44 × 10^3^	2.39 × 10^3^
F22	Min	2.30 × 10^3^	2.37 × 10^3^	2.30 × 10^3^	3.12 × 10^3^	3.39 × 10^3^	2.52 × 10^3^	2.31 × 10^3^	2.30 × 10^3^
	Std	1.69 × 10^3^	2.27 × 10^3^	9.37 × 10^2^	1.45 × 10^3^	8.31 × 10^2^	9.93 × 10^2^	1.78 × 10^3^	2.33 × 10^3^
	Mean	4.76 × 10^3^	4.97 × 10^3^	6.16 × 10^3^	7.37 × 10^3^	4.29 × 10^3^	3.71 × 10^3^	3.01 × 10^3^	4.77 × 10^3^
F23	Min	2.73 × 10^3^	2.84 × 10^3^	2.71 × 10^3^	3.01 × 10^3^	2.99 × 10^3^	2.85 × 10^3^	2.79 × 10^3^	2.72 × 10^3^
	Std	1.01 × 10^2^	7.20 × 10^1^	2.66 × 10^1^	1.30 × 10^2^	9.77 × 10^1^	5.56 × 10^1^	6.34 × 10^1^	2.39 × 10^1^
	Mean	2.91 × 10^3^	2.99 × 10^3^	2.77 × 10^3^	3.30 × 10^3^	3.20 × 10^3^	2.94 × 10^3^	2.92 × 10^3^	2.77 × 10^3^
F24	Min	2.91 × 10^3^	3.02 × 10^3^	2.87 × 10^3^	3.25 × 10^3^	3.13 × 10^3^	3.00 × 10^3^	2.95 × 10^3^	2.91 × 10^3^
	Std	1.06 × 10^2^	6.97 × 10^1^	3.91 × 10^1^	1.53 × 10^2^	1.11 × 10^2^	5.13 × 10^1^	1.19 × 10^2^	2.15 × 10^1^
	Mean	3.09 × 10^3^	3.21 × 10^3^	2.96 × 10^3^	3.50 × 10^3^	3.30 × 10^3^	3.08 × 10^3^	3.15 × 10^3^	2.95 × 10^3^
F25	Min	2.88 × 10^3^	2.90 × 10^3^	2.88 × 10^3^	2.95 × 10^3^	3.20 × 10^3^	3.00 × 10^3^	2.89 × 10^3^	2.88 × 10^3^
	Std	6.07 × 10^1^	5.33 × 10^1^	1.54 × 10^1^	3.37 × 10^1^	1.47 × 10^2^	2.00 × 10^2^	2.13 × 10^1^	1.12 × 10^1^
	Mean	2.93 × 10^3^	2.98 × 10^3^	2.90 × 10^3^	3.01 × 10^3^	3.36 × 10^3^	3.19 × 10^3^	2.93 × 10^3^	2.89 × 10^3^
F26	Min	2.80 × 10^3^	3.95 × 10^3^	4.58 × 10^3^	4.28 × 10^3^	6.90 × 10^3^	4.12 × 10^3^	3.07 × 10^3^	2.91 × 10^3^
	Std	9.35 × 10^2^	9.96 × 10^2^	3.06 × 10^2^	1.29 × 10^3^	6.01 × 10^2^	1.47 × 10^3^	1.21 × 10^3^	5.16 × 10^2^
	Mean	4.77 × 10^3^	7.04 × 10^3^	5.01 × 10^3^	8.12 × 10^3^	8.38 × 10^3^	6.69 × 10^3^	6.36 × 10^3^	4.69 × 10^3^
F27	Min	3.21 × 10^3^	3.22 × 10^3^	3.21 × 10^3^	3.23 × 10^3^	3.30 × 10^3^	3.27 × 10^3^	3.23 × 10^3^	3.20 × 10^3^
	Std	7.48 × 10^1^	1.17 × 10^2^	1.37 × 10^1^	2.26 × 10^2^	1.12 × 10^2^	8.28 × 10^1^	7.43 × 10^1^	1.93 × 10^−4^
	Mean	3.29 × 10^3^	3.40 × 10^3^	3.23 × 10^3^	3.56 × 10^3^	3.49 × 10^3^	3.40 × 10^3^	3.34 × 10^3^	3.20 × 10^3^
F28	Min	3.23 × 10^3^	3.23 × 10^3^	3.21 × 10^3^	3.36 × 10^3^	3.53 × 10^3^	3.36 × 10^3^	3.21 × 10^3^	3.26 × 10^3^
	Std	1.19 × 10^2^	4.86 × 10^2^	5.37 × 10^1^	6.91 × 10^1^	3.50 × 10^2^	3.80 × 10^2^	2.78 × 10^1^	1.01 × 10^1^
	Mean	3.35 × 10^3^	3.52 × 10^3^	3.27 × 10^3^	3.47 × 10^3^	4.24 × 10^3^	3.83 × 10^3^	3.30 × 10^3^	3.30 × 10^3^
F29	Min	3.37 × 10^3^	3.75 × 10^3^	3.59 × 10^3^	4.22 × 10^3^	5.07 × 10^3^	3.99 × 10^3^	3.72 × 10^3^	3.26 × 10^3^
	Std	2.45 × 10^2^	2.65 × 10^2^	2.69 × 10^2^	4.61 × 10^2^	4.37 × 10^2^	4.24 × 10^2^	3.12 × 10^2^	1.48 × 10^2^
	Mean	3.79 × 10^3^	4.47 × 10^3^	4.01 × 10^3^	4.94 × 10^3^	5.71 × 10^3^	4.73 × 10^3^	4.29 × 10^3^	3.51 × 10^3^
F30	Min	8.84 × 10^3^	3.24 × 10^4^	2.45 × 10^4^	7.63 × 10^5^	2.99 × 10^6^	2.09 × 10^6^	9.31 × 10^3^	3.23 × 10^3^
	Std	1.58 × 10^6^	6.06 × 10^6^	7.86 × 10^4^	1.21 × 10^7^	2.72 × 10^7^	1.60 × 10^7^	2.19 × 10^4^	5.10 × 10^3^
	Mean	4.54 × 10^5^	3.03 × 10^6^	1.10 × 10^5^	9.77 × 10^6^	3.50 × 10^7^	1.94 × 10^7^	3.77 × 10^4^	7.70 × 10^3^

**Table 3 biomimetics-09-00388-t003:** Comparison of 8 algorithms of 50 dimension.

Function		PSO	DBO	SMA	HHO	SABO	SCSO	TSO	SLTSO
F1	Min	1.20 × 10^9^	3.66 × 10^8^	2.19 × 10^6^	2.19 × 10^9^	2.49 × 10^10^	1.48 × 10^10^	3.12 × 10^8^	1.75 × 10^2^
	Std	5.83 × 10^9^	1.05 × 10^10^	2.67 × 10^6^	1.55 × 10^9^	1.04 × 10^10^	6.99 × 10^9^	5.57 × 10^8^	6.13 × 10^5^
	Mean	7.50 × 10^9^	6.57 × 10^9^	7.26 × 10^6^	5.36 × 10^9^	4.32 × 10^10^	2.77 × 10^10^	1.02 × 10^9^	1.25 × 10^5^
F3	Min	1.26 × 10^5^	1.78 × 10^5^	1.11 × 10^5^	1.41 × 10^5^	1.56 × 10^5^	9.28 × 10^4^	1.51 × 10^5^	2.45 × 10^5^
	Std	5.42 × 10^4^	6.59 × 10^4^	5.67 × 10^4^	1.87 × 10^4^	1.50 × 10^4^	2.26 × 10^4^	3.25 × 10^4^	4.52 × 10^4^
	Mean	2.22 × 10^5^	2.57 × 10^5^	2.05 × 10^5^	1.75 × 10^5^	1.83 × 10^5^	1.36 × 10^5^	2.04 × 10^5^	3.47 × 10^5^
F4	Min	5.73 × 10^2^	7.37 × 10^2^	5.16 × 10^2^	1.05 × 10^3^	3.73 × 10^3^	1.30 × 10^3^	7.22 × 10^2^	4.51 × 10^2^
	Std	5.59 × 10^2^	1.31 × 10^3^	6.20 × 10^1^	3.77 × 10^2^	2.17 × 10^3^	1.41 × 10^3^	9.90 × 10^1^	6.43 × 10^1^
	Mean	1.14 × 10^3^	1.57 × 10^3^	6.17 × 10^2^	1.69 × 10^3^	7.56 × 10^3^	4.02 × 10^3^	8.55 × 10^2^	5.55 × 10^2^
F5	Min	6.51 × 10^2^	7.48 × 10^2^	7.30 × 10^2^	8.80 × 10^2^	1.02 × 10^3^	8.67 × 10^2^	7.63 × 10^2^	6.96 × 10^2^
	Std	4.99 × 10^1^	9.11 × 10^1^	5.50 × 10^1^	2.95 × 10^1^	4.90 × 10^1^	3.88 × 10^1^	3.73 × 10^1^	2.56 × 10^1^
	Mean	7.21 × 10^2^	9.61 × 10^2^	8.21 × 10^2^	9.33 × 10^2^	1.11 × 10^3^	9.50 × 10^2^	8.39 × 10^2^	7.42 × 10^2^
F6	Min	6.09 × 10^2^	6.34 × 10^2^	6.19 × 10^2^	6.74 × 10^2^	6.61 × 10^2^	6.57 × 10^2^	6.49 × 10^2^	6.00 × 10^2^
	Std	6.61 × 10^0^	1.13 × 10^1^	1.26 × 10^1^	3.45 × 10^0^	1.10 × 10^1^	7.62 × 10^0^	7.98 × 10^0^	2.26 × 10^0^
	Mean	6.19 × 10^2^	6.62 × 10^2^	6.49 × 10^2^	6.79 × 10^2^	6.87 × 10^2^	6.75 × 10^2^	6.63 × 10^2^	6.02 × 10^2^
F7	Min	9.60 × 10^2^	1.14 × 10^3^	1.05 × 10^3^	1.65 × 10^3^	1.51 × 10^3^	1.52 × 10^3^	1.26 × 10^3^	9.87 × 10^2^
	Std	9.28 × 10^1^	1.54 × 10^2^	8.94 × 10^1^	8.43 × 10^1^	1.04 × 10^2^	1.03 × 10^2^	1.35 × 10^2^	3.87 × 10^1^
	Mean	1.09 × 10^3^	1.40 × 10^3^	1.17 × 10^3^	1.89 × 10^3^	1.66 × 10^3^	1.72 × 10^3^	1.62 × 10^3^	1.06 × 10^3^
F8	Min	9.46 × 10^2^	1.06 × 10^3^	9.58 × 10^2^	1.15 × 10^3^	1.32 × 10^3^	1.15 × 10^3^	1.01 × 10^3^	9.80 × 10^2^
	Std	4.78 × 10^1^	1.16 × 10^2^	5.33 × 10^1^	3.68 × 10^1^	5.70 × 10^1^	4.93 × 10^1^	4.53 × 10^1^	2.79 × 10^1^
	Mean	1.02 × 10^3^	1.28 × 10^3^	1.09 × 10^3^	1.23 × 10^3^	1.45 × 10^3^	1.30 × 10^3^	1.12 × 10^3^	1.05 × 10^3^
F9	Min	1.67 × 10^3^	1.10 × 10^4^	7.93 × 10^3^	2.63 × 10^4^	1.82 × 10^4^	1.40 × 10^4^	7.24 × 10^3^	1.57 × 10^3^
	Std	8.27 × 10^3^	6.69 × 10^3^	3.85 × 10^3^	2.70 × 10^3^	4.68 × 10^3^	4.00 × 10^3^	4.32 × 10^3^	3.09 × 10^3^
	Mean	1.11 × 10^4^	3.05 × 10^4^	1.58 × 10^4^	3.16 × 10^4^	3.14 × 10^4^	2.17 × 10^4^	1.39 × 10^4^	7.42 × 10^3^
F10	Min	6.32 × 10^3^	7.54 × 10^3^	6.63 × 10^3^	8.83 × 10^3^	1.41 × 10^4^	8.43 × 10^3^	6.91 × 10^3^	8.27 × 10^3^
	Std	1.02 × 10^3^	2.51 × 10^3^	8.59 × 10^2^	9.50 × 10^2^	5.07 × 10^2^	9.19 × 10^2^	1.77 × 10^3^	5.71 × 10^2^
	Mean	8.17 × 10^3^	1.24 × 10^4^	8.44 × 10^3^	1.04 × 10^4^	1.50 × 10^4^	1.04 × 10^4^	1.03 × 10^4^	9.19 × 10^3^
F11	Min	1.37 × 10^3^	2.00 × 10^3^	1.29 × 10^3^	1.84 × 10^3^	7.04 × 10^3^	3.15 × 10^3^	1.45 × 10^3^	1.30 × 10^3^
	Std	3.72 × 10^2^	4.68 × 10^3^	8.00 × 10^1^	5.80 × 10^2^	2.42 × 10^3^	2.85 × 10^3^	2.12 × 10^2^	2.50 × 10^2^
	Mean	1.72 × 10^3^	5.77 × 10^3^	1.43 × 10^3^	3.02 × 10^3^	1.12 × 10^4^	7.89 × 10^3^	1.75 × 10^3^	1.59 × 10^3^
F12	Min	1.59 × 10^7^	2.63 × 10^8^	1.05 × 10^7^	3.63 × 10^8^	3.47 × 10^9^	3.74 × 10^8^	1.24 × 10^7^	2.16 × 10^6^
	Std	2.73 × 10^9^	7.38 × 10^8^	1.73 × 10^7^	5.56 × 10^8^	5.06 × 10^9^	3.45 × 10^9^	3.18 × 10^7^	1.58 × 10^7^
	Mean	2.58 × 10^9^	1.16 × 10^9^	3.66 × 10^7^	8.49 × 10^8^	1.09 × 10^10^	4.41 × 10^9^	4.90 × 10^7^	1.95 × 10^7^
F13	Min	4.15 × 10^4^	2.37 × 10^5^	6.65 × 10^4^	6.35 × 10^6^	3.39 × 10^8^	1.37 × 10^7^	1.53 × 10^4^	2.11 × 10^3^
	Std	1.61 × 10^9^	1.18 × 10^8^	7.42 × 10^4^	3.22 × 10^7^	1.85 × 10^9^	4.70 × 10^8^	3.80 × 10^4^	7.69 × 10^3^
	Mean	9.04 × 10^8^	1.08 × 10^8^	1.79 × 10^5^	2.59 × 10^7^	2.05 × 10^9^	5.38 × 10^8^	6.98 × 10^4^	1.00 × 10^4^
F14	Min	2.23 × 10^5^	3.18 × 10^5^	1.51 × 10^5^	3.63 × 10^5^	3.65 × 10^5^	9.14 × 10^4^	1.21 × 10^4^	1.32 × 10^5^
	Std	6.21 × 10^6^	4.03 × 10^6^	6.75 × 10^5^	4.51 × 10^6^	6.59 × 10^6^	5.48 × 10^6^	2.59 × 10^5^	3.75 × 10^5^
	Mean	2.30 × 10^6^	4.03 × 10^6^	9.15 × 10^5^	4.92 × 10^6^	7.15 × 10^6^	3.41 × 10^6^	2.71 × 10^5^	6.21 × 10^5^
F15	Min	3.80 × 10^3^	3.86 × 10^4^	1.18 × 10^4^	4.18 × 10^5^	5.74 × 10^6^	3.54 × 10^4^	4.07 × 10^3^	1.71 × 10^3^
	Std	1.30 × 10^7^	2.30 × 10^8^	2.79 × 10^4^	8.83 × 10^5^	2.47 × 10^8^	5.61 × 10^8^	8.91 × 10^3^	3.99 × 10^4^
	Mean	2.41 × 10^6^	6.15 × 10^7^	4.73 × 10^4^	1.37 × 10^6^	2.06 × 10^8^	3.44 × 10^8^	1.50 × 10^4^	2.32 × 10^4^
F16	Min	2.46 × 10^3^	3.44 × 10^3^	2.67 × 10^3^	4.01 × 10^3^	4.81 × 10^3^	3.50 × 10^3^	3.20 × 10^3^	2.98 × 10^3^
	Std	5.04 × 10^2^	6.49 × 10^2^	4.64 × 10^2^	6.60 × 10^2^	5.06 × 10^2^	5.01 × 10^2^	3.77 × 10^2^	3.26 × 10^2^
	Mean	3.55 × 10^3^	4.84 × 10^3^	3.59 × 10^3^	4.90 × 10^3^	5.95 × 10^3^	4.68 × 10^3^	3.93 × 10^3^	3.83 × 10^3^
F17	Min	2.61 × 10^3^	3.31 × 10^3^	2.44 × 10^3^	3.34 × 10^3^	3.63 × 10^3^	3.14 × 10^3^	2.86 × 10^3^	2.89 × 10^3^
	Std	3.78 × 10^2^	5.32 × 10^2^	4.11 × 10^2^	4.29 × 10^2^	5.37 × 10^2^	7.09 × 10^2^	2.95 × 10^2^	2.29 × 10^2^
	Mean	3.32 × 10^3^	4.20 × 10^3^	3.37 × 10^3^	3.90 × 10^3^	4.73 × 10^3^	4.00 × 10^3^	3.45 × 10^3^	3.54 × 10^3^
F18	Min	3.31 × 10^5^	3.82 × 10^5^	1.04 × 10^6^	1.54 × 10^6^	2.95 × 10^6^	3.06 × 10^5^	1.59 × 10^5^	9.43 × 10^5^
	Std	7.38 × 10^6^	9.51 × 10^6^	4.22 × 10^6^	1.09 × 10^7^	2.47 × 10^7^	1.52 × 10^7^	1.60 × 10^6^	4.91 × 10^6^
	Mean	6.70 × 10^6^	9.41 × 10^6^	7.12 × 10^6^	1.19 × 10^7^	3.73 × 10^7^	1.57 × 10^7^	1.96 × 10^6^	5.78 × 10^6^
F19	Min	1.15 × 10^4^	6.83 × 10^4^	3.65 × 10^3^	3.07 × 10^5^	7.71 × 10^6^	1.49 × 10^5^	4.46 × 10^3^	2.04 × 10^3^
	Std	3.16 × 10^6^	7.56 × 10^6^	1.71 × 10^4^	2.35 × 10^6^	6.45 × 10^7^	2.16 × 10^7^	1.34 × 10^4^	1.61 × 10^4^
	Mean	1.56 × 10^6^	7.93 × 10^6^	2.63 × 10^4^	2.35 × 10^6^	6.26 × 10^7^	1.23 × 10^7^	2.52 × 10^4^	1.90 × 10^4^
F20	Min	2.24 × 10^3^	3.04 × 10^3^	2.95 × 10^3^	2.88 × 10^3^	3.61 × 10^3^	2.83 × 10^3^	2.60 × 10^3^	2.97 × 10^3^
	Std	4.11 × 10^2^	3.48 × 10^2^	2.89 × 10^2^	3.21 × 10^2^	2.21 × 10^2^	3.17 × 10^2^	3.70 × 10^2^	1.78 × 10^2^
	Mean	3.14 × 10^3^	3.79 × 10^3^	3.42 × 10^3^	3.57 × 10^3^	4.24 × 10^3^	3.60 × 10^3^	3.35 × 10^3^	3.44 × 10^3^
F21	Min	2.44 × 10^3^	2.66 × 10^3^	2.48 × 10^3^	2.77 × 10^3^	2.84 × 10^3^	2.67 × 10^3^	2.53 × 10^3^	2.48 × 10^3^
	Std	5.72 × 10^1^	8.41 × 10^1^	5.54 × 10^1^	8.53 × 10^1^	4.58 × 10^1^	6.23 × 10^1^	6.01 × 10^1^	3.47 × 10^1^
	Mean	2.54 × 10^3^	2.88 × 10^3^	2.57 × 10^3^	2.94 × 10^3^	2.94 × 10^3^	2.79 × 10^3^	2.65 × 10^3^	2.56 × 10^3^
F22	Min	3.94 × 10^3^	9.20 × 10^3^	7.75 × 10^3^	9.98 × 10^3^	1.54 × 10^4^	1.01 × 10^4^	8.64 × 10^3^	9.57 × 10^3^
	Std	1.92 × 10^3^	2.28 × 10^3^	9.69 × 10^2^	8.65 × 10^2^	4.95 × 10^2^	1.17 × 10^3^	1.55 × 10^3^	7.37 × 10^2^
	Mean	1.04 × 10^4^	1.25 × 10^4^	9.48 × 10^3^	1.25 × 10^4^	1.71 × 10^4^	1.23 × 10^4^	1.13 × 10^4^	1.10 × 10^4^
F23	Min	3.11 × 10^3^	3.35 × 10^3^	2.92 × 10^3^	3.69 × 10^3^	3.61 × 10^3^	3.20 × 10^3^	3.12 × 10^3^	2.89 × 10^3^
	Std	1.70 × 10^2^	1.37 × 10^2^	5.72 × 10^1^	2.12 × 10^2^	1.85 × 10^2^	1.06 × 10^2^	1.83 × 10^2^	5.64 × 10^1^
	Mean	3.35 × 10^3^	3.57 × 10^3^	3.04 × 10^3^	4.01 × 10^3^	3.91 × 10^3^	3.39 × 10^3^	3.43 × 10^3^	3.01 × 10^3^
F24	Min	3.27 × 10^3^	3.46 × 10^3^	3.02 × 10^3^	3.96 × 10^3^	3.61 × 10^3^	3.23 × 10^3^	3.39 × 10^3^	3.11 × 10^3^
	Std	1.96 × 10^2^	1.24 × 10^2^	9.09 × 10^1^	2.16 × 10^2^	1.59 × 10^2^	1.25 × 10^2^	3.11 × 10^2^	5.05 × 10^1^
	Mean	3.53 × 10^3^	3.67 × 10^3^	3.19 × 10^3^	4.31 × 10^3^	3.93 × 10^3^	3.54 × 10^3^	3.78 × 10^3^	3.24 × 10^3^
F25	Min	3.08 × 10^3^	3.16 × 10^3^	3.04 × 10^3^	3.34 × 10^3^	4.26 × 10^3^	4.26 × 10^3^	3.14 × 10^3^	2.94 × 10^3^
	Std	1.12 × 10^2^	1.74 × 10^3^	3.88 × 10^1^	2.23 × 10^2^	1.15 × 10^3^	5.79 × 10^2^	9.71 × 10^1^	4.86 × 10^1^
	Mean	3.23 × 10^3^	3.79 × 10^3^	3.10 × 10^3^	3.79 × 10^3^	6.96 × 10^3^	5.15 × 10^3^	3.32 × 10^3^	3.02 × 10^3^
F26	Min	3.71 × 10^3^	8.28 × 10^3^	2.94 × 10^3^	7.59 × 10^3^	1.10 × 10^4^	6.36 × 10^3^	8.07 × 10^3^	5.33 × 10^3^
	Std	1.66 × 10^3^	1.35 × 10^3^	2.28 × 10^3^	1.39 × 10^3^	9.68 × 10^2^	2.24 × 10^3^	1.32 × 10^3^	5.11 × 10^2^
	Mean	7.15 × 10^3^	1.10 × 10^4^	5.42 × 10^3^	1.18 × 10^4^	1.36 × 10^4^	1.08 × 10^4^	1.11 × 10^4^	6.64 × 10^3^
F27	Min	3.45 × 10^3^	3.60 × 10^3^	3.32 × 10^3^	3.87 × 10^3^	3.94 × 10^3^	3.73 × 10^3^	3.58 × 10^3^	3.20 × 10^3^
	Std	2.45 × 10^2^	2.42 × 10^2^	1.05 × 10^2^	7.13 × 10^2^	4.89 × 10^2^	2.87 × 10^2^	2.39 × 10^2^	1.78 × 10^−4^
	Mean	3.74 × 10^3^	4.01 × 10^3^	3.54 × 10^3^	4.94 × 10^3^	4.74 × 10^3^	4.31 × 10^3^	3.95 × 10^3^	3.20 × 10^3^
F28	Min	3.42 × 10^3^	3.52 × 10^3^	3.32 × 10^3^	3.99 × 10^3^	6.01 × 10^3^	4.30 × 10^3^	3.49 × 10^3^	3.30 × 10^3^
	Std	1.04 × 10^3^	2.49 × 10^3^	4.71 × 10^1^	3.91 × 10^2^	8.66 × 10^2^	6.50 × 10^2^	1.34 × 10^2^	2.05 × 10^−4^
	Mean	4.47 × 10^3^	6.67 × 10^3^	3.38 × 10^3^	4.77 × 10^3^	7.52 × 10^3^	5.55 × 10^3^	3.84 × 10^3^	3.30 × 10^3^
F29	Min	3.82 × 10^3^	4.69 × 10^3^	4.26 × 10^3^	5.69 × 10^3^	7.09 × 10^3^	5.32 × 10^3^	4.86 × 10^3^	3.99 × 10^3^
	Std	4.26 × 10^2^	8.66 × 10^2^	3.00 × 10^2^	8.53 × 10^2^	1.70 × 10^3^	8.62 × 10^2^	4.73 × 10^2^	2.11 × 10^2^
	Mean	4.66 × 10^3^	6.20 × 10^3^	4.98 × 10^3^	7.06 × 10^3^	9.83 × 10^3^	6.86 × 10^3^	5.52 × 10^3^	4.37 × 10^3^
F30	Min	1.31 × 10^6^	4.69 × 10^6^	7.83 × 10^6^	3.61 × 10^7^	2.02 × 10^8^	9.52 × 10^7^	1.46 × 10^6^	3.34 × 10^3^
	Std	8.05 × 10^6^	6.09 × 10^7^	3.82 × 10^6^	7.68 × 10^7^	2.23 × 10^8^	1.92 × 10^8^	1.69 × 10^6^	6.54 × 10^3^
	Mean	7.40 × 10^6^	6.37 × 10^7^	1.35 × 10^7^	1.38 × 10^8^	5.29 × 10^8^	2.84 × 10^8^	3.75 × 10^6^	9.58 × 10^3^

**Table 4 biomimetics-09-00388-t004:** Comparison of optimized design algorithms for welded beams.

Algorithm	*l*	*t*	*b*	*h*	*f* _min_
PSO	0.125	6.3202	8.4483	0.2661	3.4293
DBO	0.3469	3.8941	5.6022	0.5353	2.7293
SMA	0.1956	3.4175	3.4175	0.1989	1.6925
HHO	0.1708	3.7218	9.5616	0.1953	1.7859
SABO	0.2759	2.7468	7.8187	0.2889	2.5531
SLTSO	0.1981	3.3528	9.1916	0.1988	1.6713
TSO	0.1962	3.4175	9.1995	0.1989	1.6931
SCSO	0.1913	3.5671	9.1924	0.1985	1.6897

**Table 5 biomimetics-09-00388-t005:** Comparison of optimized design algorithms for gear train design.

Algorithm	*l*	*t*	*b*	*h*	*f* _min_
PSO	12	23.869	51.1725	38.7949	1.7009 × 10^−14^
DBO	16.592	12	48.651	28.3652	9.9416 × 10^−28^
SMA	12	16.3463	36.245	37.5114	2.6689 × 10^−11^
HHO	15.2728	27.4423	53.8562	53.9388	1.2646 × 10^−24^
SABO	27.5857	14.3694	49.5786	55.4162	1.4442 × 10^−11^
SLTSO	12	24.0508	38.6095	51.8098	1.9259 × 10^−32^
TSO	15.2097	34.1496	54.3547	55.1563	1.2646 × 10^−27^
SCSO	21.0187	33.6516	78.4484	62.5337	9.4023 × 10^−9^

## Data Availability

The data that support the findings of this study are available from the corresponding author upon request. There are no restrictions on data availability.

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
