# Peer review of "Path Planning of Unmanned Aerial Vehicles Based on an Improved Bio-Inspired Tuna Swarm Optimization Algorithm"

_biomimetics, 2024, doi:10.3390/biomimetics9070388_

Round 1
Reviewer 1 Report
Comments and Suggestions for Authors
This is amazing work, and there is no issue, so I directly recommend it to be published.
Comments on the Quality of English Languageok
Author Response
Thank you for arranging for the timely review of the paper and for your response.Thank you for allowing a resubmission of our manuscript, with an opportunity to address the reviewers’ comments.We are uploading our point-by-point response to the comments.

Reviewer 2 Report
Comments and Suggestions for Authors
Some of the Reviewer comments given below:
1. What is novelty this paper?
2. How does the proposed Sine Levy Tuna Swarm Optimization (SLTSO) algorithm specifically address the limitations of the existing Tuna Swarm Optimization (TSO) algorithm in terms of global search capability and avoidance of local optima?
3. Could you elaborate on how the SLTSO algorithm integrates the golden sine technique to optimize individual locations, and how this contributes to its improved ability to search globally and avoid local optima, particularly in complex obstacle environments for UAV route planning?
4. Provide more insights into the specific criteria used to evaluate the optimization performance and stability of the SLTSO algorithm compared to the other swarm intelligence algorithms across the 30 benchmark test functions from the CEC2018 suite? Additionally, how did the SLTSO algorithm perform in terms of convergence rates and solution quality on these test functions, considering the inherent issues encountered with the F2 function?
5. The paper highlights significant performance differences between the SLTSO algorithm and others, particularly for high-dimensional multi-modal functions (Functions F20 to F30). Could you elaborate on how the SLTSO algorithm effectively addresses the challenges posed by these functions compared to the other algorithms, and what specific mechanisms contribute to its superior convergence precision and speed?
6. The consistent settings of population size and iteration count across all algorithms aim to mitigate randomness in heuristic algorithms. Could you discuss how these settings were chosen, and how robust the conclusions drawn from this experimental setup are regarding the comparative performance of the SLTSO algorithm against the other algorithms across the various test functions and dimensions?
7. How the effectiveness of the SLTSO algorithm and the other seven algorithms was evaluated in the UAV path planning scenario, particularly regarding the criteria used for comparison and the specific statistical results obtained from the 30 iterations for each algorithm?
Comments on the Quality of English Languagenil
Author Response

(The authors gave the same response as above.)

Reviewer 3 Report
Comments and Suggestions for Authors
1. Similarity index is more. It is a major concern in the manuscript. It comes around 36% as per iThenticate report.
2. In abstract, the following statement “The SLTSO algorithm produces shorter and smoother pathways, demonstrates quicker convergence rates, improved optimization precision, and simultaneously lowers energy consumption, according to experimental data” should be supported with analytical data. Similarly in conclusion too.
3. In introduction the following can be included “Unmanned ground vehicle for surveillance DOI: 10.1109/ICCCNT49239.2020.9225313” as additional.
4. Introduction was clearly projected by the authors. It is clear to read and also understandable.
5. What axis represented in x y z as per equation 1?
6. What is Li in equation 6?
7. Section 6.2 Comparison of engineering application is not clear. It requires rewriting of paragraph so that the flow can be understandable.
8. Author required to clarify that why some path planning algorithms crossing objects/ obstacles as shown in Figure 9. Top view of path planning for 8 algorithms.
9. No much information were found in Figure 10. Iterative effect diagram of path planning for 8 algorithms. Either this figure can be removed or can be changed to communicate meaningful information.
Comments on the Quality of English LanguageMinor editing of English language required
Author Response

(The authors gave the same response as above.)

Reviewer 4 Report
Comments and Suggestions for Authors
This paper presents a variation of particle swarm optimization for aerial vehicle path finding. Comparison work is good.
However, there are too many hyper-parameters that are not specified in the simulation study. Also, the simulation is too simple. More rigorous stimulation study should be performed. The used UAV model is too simple. More realistic one should be considered. Hyper-parameters that are used in the simulation study must be specified and discussed their impact to the results. Also, their algorithm heavily relies on the bound of the parameteres. The robustness of such conditions should be studied.
Author Response

(The authors gave the same response as above.)

Round 2
Reviewer 4 Report
Comments and Suggestions for Authors
- Selection of hyper-paramters or their bounds such as r_g1, r_g2, a, b, \tau, C1Z(?), R_4, R_5 must be discussed further why these values are effective.
- In (19), why 3 cos (*) in the exponent?
- In (35), what is (23a), (23b), \beta?
It is hard to understand 5.3.
Author Response

(The authors gave the same response as above.)
